# Lipedema World Alliance Delphi Consensus-Based Position Paper on the Definition and Management of Lipedema: Results from the 2023 Lipedema World Congress in Potsdam

Philipp Kruppa [1] ✉, Rachelle Crescenzi [2,3], Gabriele Faerber [4],
Isabel Forner-Cordero [5], Manuel Cornely [6], Ramin Shayan [7], Tara Karnezis[7],
Jose Luis Simarro [8], Paula Frederichi de Souza[9], Karen Louise Herbst [10],
Mojtaba Ghods [1,12] & Sandro Michelini [11]

Lipedema predominantly affects women and is characterized by an abnormal distribution of adipose tissue, accompanied by pain or discomfort in affected areas. Despite growing awareness, inconsistent diagnostic criteria and treatment approaches hinder medical care and research. This multi-phase Delphi study was conducted to address the need for internationally accepted consensus on fundamental aspects of the disease. Through online surveys and an in-person discussions, experts representing 19 countries evaluated on 62 original statements regarding (1) clarity, (2) agreement, (3) recommendation for inclusion, (4) strength of evidence, and (5) whether additional evidence was needed. Ultimately, 59 statements reached consensus across eight domains encompassing the definition and management of lipedema. The findings provide a framework to guide internationally applicable recommendations for patients with lipedema that may improve outcomes globally. Limited evidence in several areas highlights the importance of further research, standardization of data reporting, and international collaboration among healthcare providers, researchers, and patient advocates to address this women's health disparity effectively.

Lipedema was first described in the 1940s by Drs. Allen and Hines at the Mayo Clinic, who identified it as a clinical syndrome characterized by abnormal and symmetrical accumulation of fat in the lower extremities of women, frequently accompanied by physical ache, feet sparing orthostatic edema, and psychological distress[1]. For decades, lipedema was not widely recognized and was often confused with obesity or lymphedema, leading to misdiagnosis and inadequate treatment[2]. Depending on the availability of healthcare professionals and services for affected individuals within various healthcare systems, lipedema remains underdiagnosed today, with a limited

[1]Department of Plastic, Aesthetic and Reconstructive Microsurgery/Hand surgery, Hospital Ernst von Bergmann, Potsdam, Germany. [2]Department of Radiology and Radiological Sciences, Department of Biomedical Engineering, Vanderbilt University Medical Center, Nashville, Tennessee, USA. [3]Department of Radiology and Medical Imaging, Department of Biomedical Engineering, University of Virginia, Charlottesville, Virginia, USA. [4]Centre for Vascular Medicine, Hamburg, Germany. [5]Lymphedema Unit, Rehabilitation Department, Hospital Universitari i Politècnic La Fe, University of Valencia, Valencia, Spain. [6]LY. SEARCH gGmbH, Institute for Lymphological Basic Research, Düsseldorf, Germany. [7]Lymphatic, Adipose and Regenerative Medicine Laboratory, Department of O'Brien Institute, St Vincent's Institute of Medical Research, Fitzroy, Melbourne, Australia. [8]The Lipedema Institute, Madrid, Spain. [9]Clinic Paula Frederichi, Bela Vista, São Paulo, Brazil. [10]The Roxbury Institute, Beverly Hills, CA and Tucson, AZ, USA. [11]Vascular Diagnostics and Rehabilitation Service, San Giovanni Battista Hospital Rome and San Giuseppe Hospital Marino (Rome), ASL Roma 6, Roma, Italy. [12]Faculty of Health Sciences Brandenburg (FGW), University of Potsdam, Potsdam, Deutschland. ✉e-mail: kruppaph@gmail.com

understanding continuing within the medical community. In recent years, however, awareness of lipedema as a distinct clinical entity has increased. This improved disease recognition has been driven by a combination of patient advocacy, research, and clinical interest, prompting the development of national guidelines and consensus documents aimed at improving diagnosis and management[3–10]. Despite these advances, however, variability in diagnostic criteria and treatment approaches persists. The lack of standardized diagnostic criteria not only hinders early detection but also complicates research efforts and the development of evidence-based therapies[11]. There is therefore an urgent need for a unified, internationally accepted consensus on the fundamental aspects of the disease entity and management approaches to caring for patients with lipedema.

This multi-step Delphi consensus study was conducted to address the need for a standardized definition and management approaches to lipedema. By leveraging the collective expertise of clinicians, researchers, and patient representatives from around the world, 59 statements were ultimately recommended for inclusion and reached consensus across eight domains: 36 reached 90–100% agreement, 17 reached 80–90% agreement, and 6 reached 70% agreement. This position paper aims to advance consensus on a standardized, globally relevant definition of lipedema and to delineate principles for its evidence-based management, integrating both empirical data and expert clinical insight. In addition, where evidence was found to be insufficient, further research is called for to improve the lives of individuals with lipedema.

## Results
### Overall results of Delphi process
After three online Delphi rounds, 59 statements on the definition and management of lipedema reached consensus according to the pre-defined 70 percent agreement threshold on criterion (ii) and (iii). Thirty-six statements reached 90–100% agreement, seventeen statements reached 80–90% agreement, and six statements reached 70% agreement (Table 1).

In the final Delphi evaluation round 3, 71 out of 103 founding members of the LWA participated (34 female, 37 male), representing 19 countries across 5 continents (Table 2). The majority of respondents reported having extensive experience in working with patients with lipedema (Table 3). The professional backgrounds of the participants were diverse, encompassing medical management, conservative and surgical treatment providers, as well as researchers and patient advocates (Table 4). Patient advocates from seven countries participated, with most (ten of twelve) originating from European nations.

Domain 1: Definition & Leading Symptoms

**Statement 1:.** "**Lipedema is a chronic disease**"

| | |
|---|---|
| Clarity of statement | 100.0% |
| **Level of agreement** | **94.4%** |
| Inclusion Rating | 93.0% |
| Strength of Evidence | 72.9% |
| Additional Evidence Needed | 28.2% |

Expert Panel Comment:

The definition of 'chronic' disease varies but typically encompasses conditions that (1) have complex etiology, (2) persist for three months or longer, (3) require ongoing medical management, and (4) may be associated with functional impairment or significantly impact daily activities[12]. Patients with lipedema typically report long courses of the disease and symptoms that last many years before diagnosis or initiation of appropriate therapeutic measures[9,10,13–16]. However, systematic longitudinal studies on the chronicity of the disease are not yet available.

**Statement 2:.** "**Untreated lipedema typically presents as symmetrical, bilateral enlargement of subcutaneous adipose tissue in the extremities, accompanied by pain and/or discomfort**".

| | |
|---|---|
| Clarity of statement | 90,0% |
| **Level of agreement** | **88,6%** |
| Inclusion Rating | 91,3% |
| Strength of Evidence | 76,8% |
| Additional Evidence Needed | 33,8% |

Expert Panel Comment:

The characteristic presentation of lipedema as symmetrical and bilateral enlargement of subcutaneous adipose tissue (SAT) in the extremities is well-documented in the clinical literature[1,17,18]. The existing literature consistently describes the unique distribution of SAT predominantly in the legs, typically sparing the feet[1], and contributing to the characteristic "column-like" appearance. However, it is important to note that variations in clinical presentation exist, and that some patients may demonstrate asymmetry as the disease progresses or in the presence of comorbidities[16,19].

Pain in lipedema is a common and often debilitating symptom, significantly impacting patients' quality of life. The presence of pain varies widely among individuals, ranging from mild discomfort to severe, chronic pain. It typically manifests as a dull, aching sensation in the affected areas, exacerbated by prolonged standing or walking. In addition, patients may experience tenderness to touch and increased sensitivity in the affected regions[20–25]. More research into the nature of pain and discomfort in lipedema is needed to support disease diagnosis and mechanisms for potential interventions.

**Statement 3:.** "**Lipedema is characterized by a disproportional expansion of the subcutaneous adipose tissue of the extremities compared to that in the torso**".

| | |
|---|---|
| Clarity of statement | 91,0% |
| **Level of agreement** | **89,6%** |
| Inclusion Rating | 89,6% |
| Strength of Evidence | 71,9% |
| Additional Evidence Needed | 37,9% |

Expert Panel Comment:

In the absence of comorbid conditions, the characteristic lipedema phenotype is recognized for its hallmark feature of disproportionate fat accumulation in the SAT of the extremities compared to that in the torso. Imaging studies have consistently demonstrated increased fat deposition in the legs, supporting the assertion of disproportionate adipose tissue expansion[26–32]. However, inconsistent terminology for anatomical regions in the literature necessitates clarification in order to more clearly discuss the regions affected by lipedema. In this context, the extremities include the shoulder girdle (Cingulum membri superioris) together with the free upper limbs (Pars libera membri superioris) and the pelvic girdle (Cingulum membri inferioris) together with the free lower limbs (Pars libera membri inferioris). The torso or trunk in this context refers to the central part of the body, including the head and neck region, thorax, abdomen and pelvis[33,34].

**Statement 4:.** "**Lipedema can involve excess adipose tissue deposition in the upper extremities in a symmetrical and bilateral distribution**".

| | |
|---|---|
| Clarity of statement | 94,1% |
| **Level of agreement** | **94,0%** |
| Inclusion Rating | 89,4% |
| Strength of Evidence | 67,6% |
| Additional Evidence Needed | 28,4% |

**Table 1 | Summary of statements**

| No. | Statement | Clarity of statement (%) | Level of agreement (%) | Inclusion Rating (%) | Strength of Evidence (%) | Additional Evidence Needed (%) |
|---|---|---|---|---|---|---|
| DOMAIN 1: Definition & leading symptoms | | | | | | |
| 1 | Lipedema is a chronic disease. | 100,00 | 95,77 | 94,37 | 72,86 | 28,17 |
| 2 | Untreated lipedema typically presents as symmetrical, bilateral enlargement of subcutaneous adipose tissue in the extremities, accompanied by pain and/or discomfort. | 91,43 | 90,00 | 91,30 | 78,26 | 32,35 |
| 3 | Lipedema is characterized by a disproportional expansion of the subcutaneous adipose tissue of the extremities compared to that in the torso. | 91,04 | 89,55 | 89,55 | 71,88 | 37,88 |
| 4 | Lipedema can involve excess adipose tissue deposition in the upper extremities in a symmetrical and bilateral distribution. | 94,12 | 94,03 | 89,39 | 67,65 | 28,36 |
| 5 | Lipedema typically spares hands and feet of excess fat deposition. | 96,97 | 90,77 | 92,42 | 74,63 | 29,69 |
| 6 | Physical sensitivity to pressure and/or stretch is observed by methods such as palpation, and is mainly reported by patients as pain. | 100,00 | 98,51 | 98,48 | 59,70 | 38,81 |
| 7 | Increased sensitivity and pain caused by lipedema seem to be restricted to the body areas with lipedema-related volume increase. | 92,65 | 86,76 | 85,07 | 56,72 | 37,88 |
| 8 | Patients often report swelling or heaviness in affected areas. | 92,65 | 89,71 | 88,24 | 54,41 | 36,76 |
| 9 | Pitting edema is usually not present in lipedema-affected tissue. | 91,18 | 91,18 | 88,24 | 51,52 | 42,65 |
| 10 | Patients with lipedema often experience easy bruising in affected areas. | 95,45 | 94,03 | 92,54 | 58,21 | 35,82 |
| 11 | Kaposi-Stemmer's sign is usually negative in lipedema. | 93,94 | 95,52 | 95,38 | 71,64 | 22,73 |
| DOMAIN 2: Pathophysiology | | | | | | |
| 12 | Lipedema is a disease involving subcutaneous adipose tissue. | 90,91 | 92,54 | 90,91 | 78,79 | 20,90 |
| 13 | Numerous findings suggest that inflammation may contribute to the pathogenesis of lipedema. | 95,16 | 87,69 | 83,08 | 49,21 | 60,94 |
| 14 | Numerous findings suggest that hormonal factors may contribute to the pathogenesis of lipedema. | 96,83 | 96,88 | 92,06 | 58,06 | 56,25 |
| 15 | Several findings suggest that extracellular fluid volume might be elevated in lipedema-affected tissue compared to BMI-matched unaffected controls. | 87,10 | 76,19 | 72,58 | 46,77 | 66,13 |
| DOMAIN 3: Epidemiology | | | | | | |
| 16 | Lipedema primarily affects biological females. Occurrence in biological males appears to be possible but rare. | 96,72 | 88,71 | 88,89 | 69,84 | 33,87 |
| 17 | Hormonal changes may trigger or exacerbate the symptoms of lipedema. | 100,00 | 98,39 | 98,36 | 63,93 | 36,51 |
| 18 | Lipedema is hereditary in some cases. | 96,77 | 88,71 | 88,89 | 63,93 | 47,54 |
| 19 | The prevalence of lipedema in the adult female population remains unknown. Estimates range from less than 1% to up to 12%. | 96,83 | 79,03 | 83,87 | 37,10 | 64,52 |
| DOMAIN 4: Comorbidities & concomitant diseases | | | | | | |
| 20 | Obesity is a frequently observed concomitant disease in patients with lipedema. | 98,41 | 88,89 | 92,06 | 62,90 | 44,44 |
| 21 | Lipedema is not an obesity-related comorbidity. | 98,39 | 92,06 | 91,80 | 67,74 | 32,79 |
| 22 | Body mass index (BMI) has limited value in distinguishing between lipedema and obesity. Therefore, it is advisable to utilize the Waist-to-Height Ratio (WHtR) to exclude or assess obesity. | 95,16 | 77,42 | 78,69 | 64,52 | 40,98 |
| 23 | In cases where lipedema coincides with obesity, lipedema symptoms can be expected to persist after bariatric surgery. | 91,94 | 88,89 | 87,10 | 58,73 | 44,26 |
| 24 | Concomitant lymphostasis can develop in lipedema. | 90,16 | 88,52 | 86,67 | 44,26 | 53,33 |
| 25 | Several findings suggest that the prevalence of hypothyroidism might be higher in lipedema patients than in non-lipedema populations with comparable BMI and age. | 93,44 | 72,88 | 68,85 | 38,33 | 72,13 |
| 26 | | 96,72 | 75,41 | 72,13 | 36,67 | 66,67 |

**Table 1 (continued) | Summary of statements**

| No. | Statement | Clarity of statement (%) | Level of agreement (%) | Inclusion Rating (%) | Strength of Evidence (%) | Additional Evidence Needed (%) |
|---|---|---|---|---|---|---|
| | Lipedema might be associated with connective tissue disorders, such as hypermobility spectrum disorders. | | | | | |
| DOMAIN 5: Impact on quality of life and symptom burden | | | | | | |
| 27 | Lipedema can negatively impact mental health and overall quality of life. | 100,00 | 98,36 | 98,33 | 70,00 | 39,34 |
| 28 | If present, psychological involvement may be caused by lipedema-related symptoms rather than being the cause of those symptoms. | 93,33 | 90,00 | 85,25 | 47,54 | 54,10 |
| 29 | Missed or delayed diagnosis or management of lipedema negatively affects a patient's symptom burden, mental well-being, and overall quality of life. | 100,00 | 100,00 | 100,00 | 67,21 | 31,67 |
| 30 | Missed or delayed diagnosis or management of lipedema increases the cost burden for patients and the healthcare system. | 100,00 | 93,22 | 91,67 | 56,67 | 50,00 |
| DOMAIN 6: Diagnostic approach | | | | | | |
| 31 | The clinical diagnosis of lipedema relies on the patient's medical history, physical examination and exclusion of differential diagnoses. | 98,36 | 98,36 | 98,36 | 71,67 | 35,59 |
| 32 | Currently, no imaging, serological or genetic tests, or clinical measurement instruments, are officially approved to verify the clinical diagnosis. | 98,36 | 93,44 | 93,33 | 67,21 | 38,33 |
| 33 | Routine clinical exams should include standardized anthropometric measurements, such as waist-to-height ratio (WHtR), waist-to-hip ratio (WHR), and body mass index (BMI). | 96,67 | 81,97 | 75,41 | 56,67 | 44,07 |
| 34 | The clinical classification of lipedema into stages does not reflect the complete symptom severity. | 96,67 | 95,00 | 93,33 | 60,00 | 53,33 |
| 35 | The current clinical classification for lipedema into stages has limited relevance for the disease management. | 100,00 | 91,80 | 90,16 | 53,33 | 50,00 |
| 36 | A progression in the severity of lipedema-associated symptoms depends on various factors and is not universal. | 93,33 | 95,00 | 95,00 | 46,67 | 58,33 |
| 37 | The excess limb volume in lipedema is generally not associated with obesity. | 93,10 | 81,03 | 77,59 | 39,66 | 51,67 |
| 38 | The clinical classifications based on localization have only descriptive significance. | 95,00 | 88,33 | 86,67 | 38,33 | 44,07 |
| DOMAIN 7: Treatment modalities | | | | | | |
| 39 | All therapeutic interventions of lipedema aim at alleviating symptoms and preventing or delaying progression. | 95,00 | 90,00 | 90,00 | 55,00 | 46,55 |
| 40 | Comprehensive disease management requires a multidisciplinary approach tailored to individual needs, which may involve physicians, physical therapists, dietitians, and mental health professionals. | 96,67 | 96,67 | 96,61 | 63,33 | 42,37 |
| 41 | Lipedema pain and physical sensitivity in lipedema-affected areas have been reported to be reduced by bandaging, compression, complete physical decongestive therapy or other physical therapies (such as shock wave therapy), dietary changes, tailored exercise, and lipedema reduction surgery, with varying effect sizes and durations. | 86,44 | 86,67 | 84,75 | 46,67 | 66,67 |
| 42 | Conservative management of lipedema should include lifestyle and nutritional optimization, compression therapy, and exercise to alleviate symptoms and improve quality of life. | 95,00 | 93,33 | 89,83 | 64,41 | 43,10 |
| 43 | Active self-management can help control lipedema-related symptoms and improve the overall quality of life. | 96,67 | 93,22 | 91,53 | 36,67 | 58,62 |
| 44 | Although Complex (also known as Complete) Decongestive Therapy (CDT) can be an important and effective treatment even for early-stage lipedema, not all components are required for every patient. | 95,00 | 86,67 | 86,67 | 40,00 | 52,54 |
| 45 | Nutritional guidance can help patients manage their weight, optimize overall health, reduce lipedema-associated symptoms and improve their response to therapeutic interventions. | 96,67 | 93,33 | 93,33 | 66,67 | 48,28 |
| 46 | Although pathological subcutaneous adipose tissue in lipedema is known to be largely resistant to dietary interventions, addressing overall weight loss in coincident obesity may result in symptom improvement. | 94,92 | 93,22 | 93,10 | 61,40 | 41,38 |

**Table 1 (continued) | Summary of statements**

| No. | Statement | Clarity of statement (%) | Level of agreement (%) | Inclusion Rating (%) | Strength of Evidence (%) | Additional Evidence Needed (%) |
|---|---|---|---|---|---|---|
| 47 | Considering that obesity worsens the manifestations of lipedema, disease management should include weight optimization, with a focus on waist-to-height ratio (WHtR) and waist-to-hip ratio (WHR). | 91,67 | 84,75 | 81,67 | 51,67 | 50,85 |
| 48 | Psychological and social support, addressing body image issues, mental well-being, and coping strategies, can be important to address the symptom burden of patients living with lipedema. | 96,67 | 98,28 | 96,61 | 67,80 | 37,93 |
| 49 | Tailored exercises, such as physical activity in water, walking, and yoga, can help maintain mobility, address lipedema related symptoms and support weight management in individuals with lipedema. | 94,92 | 91,53 | 90,00 | 55,93 | 43,33 |
| 50 | Currently, there is no evidence for the effectiveness of any pharmacological interventions in treating lipedema. | 100,00 | 93,22 | 93,10 | 59,32 | 44,83 |
| 51 | In cases where lipedema coincides with obesity and metabolic disease, it is advisable to prioritize treatment for obesity before considering lipedema reduction surgery. | 100,00 | 93,22 | 91,53 | 66,10 | 41,38 |
| 52 | Lymph vessel-sparing lipedema reduction surgery should be considered when there is potential for a positive impact on lipedema-related symptoms. | 86,44 | 89,47 | 87,72 | 57,89 | 50,00 |
| 53 | Surgical interventions should be performed by healthcare providers with extensive knowledge in lipedema management, including conservative treatments, as part of an integral approach. | 98,31 | 98,31 | 98,31 | 52,54 | 43,86 |
| DOMAIN 8: Future directions | | | | | | |
| 54 | Raising awareness about lipedema within the medical community and wider society is essential to reduce misdiagnosis and stigma. | 100,00 | 98,31 | 96,61 | 66,10 | 30,51 |
| 55 | A comprehensive standardized case report form (CRF) should be developed to improve consistency in diagnosis, lipedema reporting, follow-up, and research, and to facilitate, for example, both cohort and longitudinal studies. | 100,00 | 96,61 | 94,92 | 55,17 | 43,10 |
| 56 | Further research is needed to elucidate the biological mechanisms underlying lipedema, leading to objective diagnostic criteria and targeted therapies. | 98,31 | 100,00 | 98,31 | 56,90 | 45,61 |
| 57 | Studies are required to validate diagnostic modalities for lipedema, which assess their reproducibility, sensitivity, and specificity. | 100,00 | 100,00 | 96,61 | 57,89 | 42,11 |
| 58 | Long-term studies are required to assess the efficacy and safety of treatment modalities for lipedema. | 100,00 | 100,00 | 98,31 | 57,14 | 47,27 |
| 59 | Collaborative efforts between patients, researchers, clinicians and advocacy groups are crucial for advancing knowledge. Translational practice-based application of research knowledge should improve patient care. | 100,00 | 98,31 | 94,92 | 56,14 | 36,84 |

**Table 2 | Country of origin of experts participating in the study**

| Specialty | Completed Delphi evaluation Round 1 (n = 48) | Completed Delphi evaluation Round 2 (n = 49) | Completed Delphi evaluation Round 3 (n = 71) |
|---|---|---|---|
| Australia | 0 | 1 | 2 |
| Austria | 1 | 2 | 5 |
| Belgium | 0 | 1 | 1 |
| Brazil | 1 | 1 | 2 |
| Denmark | 3 | 1 | 1 |
| Germany | 11 | 7 | 16 |
| Greece | 1 | 1 | 1 |
| Hungary | 0 | 1 | 1 |
| India | 0 | 0 | 1 |
| Italy | 6 | 10 | 10 |
| Norway | 1 | 1 | 1 |
| Poland | 2 | 2 | 2 |
| Portugal | 3 | 1 | 3 |
| Spain | 8 | 4 | 8 |
| Sweden | 3 | 3 | 1 |
| Switzerland | 0 | 0 | 1 |
| Turkey | 0 | 0 | 1 |
| United Kingdom | 4 | 5 | 7 |
| United States | 4 | 8 | 7 |

**Table 3 | Years of experience working/volunteering with patients diagnosed with lipedema**

| Specialty | Completed Delphi evaluation Round 1 (n = 48) | Completed Delphi evaluation Round 2 (n = 49) | Completed Delphi evaluation Round 3 (n = 71) |
|---|---|---|---|
| > 10 years | 26 | 25 | 41 |
| 10 years | 2 | 2 | 7 |
| 9 years | 1 | 4 | 1 |
| 8 years | 5 | 2 | 3 |
| 7 years | 1 | 1 | 4 |
| 6 years | 3 | 4 | 4 |
| 5 years | 6 | 3 | 5 |
| 4 years | 0 | 2 | 1 |
| 3 years | 2 | 3 | 2 |
| 2 years | 1 | 0 | 0 |
| N/A | 1 | 3 | 3 |

Expert Panel Comment:

The involvement of the upper extremities (as defined in the "expert panel comment" of statement 3) in lipedema has been observed in both early and advanced stages of the disease. Studies report varying percentages of lipedema patients with arm involvement, ranging from 30% to 80%[13,16,19,35].

**Statement 5:. "Lipedema typically spares hands and feet of excess fat deposition".**

| | |
|---|---|
| Clarity of statement | 97,0% |
| **Level of agreement** | **90,9%** |
| Inclusion Rating | 92,5% |
| Strength of Evidence | 73,5% |
| Additional Evidence Needed | 30,8% |

Expert Panel Comment:

According to the original publication of Allen and Hines, lipedema adipose tissue deposition "usually does not involve the feet"[1]. Recent publications defining diagnostic criteria for lipedema also confirm this for the arms and often describe a separation between normal and abnormal tissue ("Cuff") at the ankle or wrist[4–6,8–10,35]. However, although not constituting definitive evidence, patient reports suggest that lipedema-related symptoms may manifest in hands and feet as the disease progresses or in the presence of comorbidities[16,19].

**Statement 6:. "Physical sensitivity to pressure and/or stretch is observed by methods such as palpation, and is mainly reported by patients as pain".**

| | |
|---|---|
| Clarity of statement | 98,5% |
| **Level of agreement** | **97,0%** |
| Inclusion Rating | 98,5% |
| Strength of Evidence | 58,2% |
| Additional Evidence Needed | 40,3% |

Expert Panel Comment:

According to the International Association for the Study of Pain (ISAP), pain is defined as an unpleasant sensory and emotional experience associated with, or resembling that associated with, actual or potential tissue damage[36]. Pain in response to typically innocuous stimuli (such as pressure, palpation, or stretching applied to lipedema-affected tissue) is a central symptom of lipedema, occurring both superficially and subcutaneously, predominantly throughout the legs or arms. Despite its significance, the literature lacks sufficient investigation and characterization of lipedema-related pain, making it complex and challenging to define[23,25,37]. The precise pathogenesis of pain in lipedema remains uncertain, with proposed mechanisms including central sensitization, nociceptive pain, and autonomic peripheral neuropathies[20,21].

**Statement 7:. "Increased sensitivity and pain caused by lipedema seem to be restricted to the body areas with lipedema-related volume increase".**

| | |
|---|---|
| Clarity of statement | 92,6% |
| **Level of agreement** | **85,3%** |
| Inclusion Rating | 83,6% |
| Strength of Evidence | 55,2% |
| Additional Evidence Needed | 39,4% |

Expert Panel Comment:

The clinical symptoms of lipedema typically manifest in the areas affected by the enlargement of SAT[25,37–39]. As outlined in statements 2 – 5, this manifestation is typically confined to the extremities, excluding the hands and feet. However, although not constituting definitive evidence, patient reports suggest that lipedema-related symptoms may be present in other body regions as the disease progresses or in the presence of comorbidities[16].

**Statement 8:. "Patients often report swelling or heaviness in affected areas".**

| | |
|---|---|
| Clarity of statement | 92,6% |
| **Level of agreement** | **89,7%** |
| Inclusion Rating | 88,2% |
| Strength of Evidence | 54,4% |
| Additional Evidence Needed | 36,8% |

Expert Panel Comment:

Patients with lipedema frequently describe the affected areas to exert a feeling of swelling or a sensation of heaviness[40]. Incidence data regarding these symptoms are limited in their availability[16].

**Table 4 | Number of experts participating in the study by specialty**

| Specialty | Completed Delphi evaluation Round 1 (n = 48) | Completed Delphi evaluation Round 2 (n = 49) | Completed Delphi evaluation Round 3 (n = 71) |
|---|---|---|---|
| Therapist (occupational, physical, lymphedema) | 3 | 4 | 4 |
| Surgeon | 13 | 8 | 17 |
| Phlebologist | 5 | 2 | 4 |
| Patient representative | 14 | 11 | 12 |
| Other type of healthcare provider | 3 | 1 | 3 |
| Nurse practitioner or physician assistant | 1 | 2 | 3 |
| Medical researcher | 1 | 8 | 7 |
| Medical doctor (non-surgeon) | 2 | 5 | 5 |
| Lymphologist | 3 | 4 | 7 |
| Gynecologist | 0 | 0 | 1 |
| General practitioner | 0 | 0 | 1 |
| Dermatologist | 1 | 2 | 2 |
| Angiologist | 2 | 2 | 5 |

**Statement 9:. "Pitting edema is usually not present in lipedema-affected tissue".**

| | |
|---|---|
| Clarity of statement | 91,2% |
| **Level of agreement** | **89,7%** |
| Inclusion Rating | 86,8% |
| Strength of Evidence | 50,0% |
| Additional Evidence Needed | 44,1% |

Expert Panel Comment:

Existing clinical evidence supports the assertion that, in the absence of comorbid conditions, pitting edema is generally absent in lipedema-affected tissue. In this regard, lipedema differs significantly from lymphedema[41]. As in the rest of the population, some individuals with lipedema may present with lymphedema, however, it is unknown whether this secondary condition represents primary or secondary lymphedema. Regardless, a causative role in lipedema has not been established.

**Statement 10:. "Patients with lipedema often experience easy bruising in affected areas".**

| | |
|---|---|
| Clarity of statement | 95,5% |
| **Level of agreement** | **94,0%** |
| Inclusion Rating | 92,5% |
| Strength of Evidence | 58,2% |
| Additional Evidence Needed | 35,8% |

Expert Panel Comment:

A frequently reported symptom of patients with lipedema is a tendency to bruise without recollection of preceding trauma within the affected areas, without bleeding from other sites, and in the absence of systemic disorders known to result in bruising. Availability of incidence data is limited, with up to 90.6% of patients reporting easy bruising.[16,40,42,43]. However, it is important to note that easy bruising is not pathognomonic for lipedema and can have various etiologies.

**Statement 11:. "Kaposi-Stemmer's sign is usually negative in lipedema".**

| | |
|---|---|
| Clarity of statement | 93,9% |
| **Level of agreement** | **95,5%** |
| Inclusion Rating | 95,4% |
| Strength of Evidence | 71,6% |
| Additional Evidence Needed | 22,7% |

Expert Panel Comment:

Kaposi-Stemmer's sign, a clinical indicator of lymphedema characterized by the inability to pinch a fold of skin at the base of the second toe or second finger[44,45], is typically considered negative in lipedema[46]. If there is lymphostasis in addition to lipedema, the sign may be positive.

Domain 2: Pathophysiology

**Statement 12:. "Lipedema is a disease involving subcutaneous adipose tissue".**

| | |
|---|---|
| Clarity of statement | 92,4% |
| **Level of agreement** | **94,0%** |
| Inclusion Rating | 90,9% |
| Strength of Evidence | 78,8% |
| Additional Evidence Needed | 19,4% |

Expert Panel Comment:

Adipose tissue, commonly known as fat, is a type of connective tissue that consists of lipid-filled cells (adipocytes) surrounded by a matrix of collagen fibers, blood vessels, fibroblasts, and immune cells[47]. Lipedema has been reported to affect components of the SAT[32,38,39,48–56].

**Statement 13:. "Numerous findings suggest that inflammation may contribute to the pathogenesis of lipedema".**

| | |
|---|---|
| Clarity of statement | 95,2% |
| **Level of agreement** | **87,7%** |
| Inclusion Rating | 83,1% |
| Strength of Evidence | 49,2% |
| Additional Evidence Needed | 60,9% |

Expert Panel Comment:

Studies have proposed that inflammatory processes may play a role in the development and progression of lipedema[39,48,49,51,53,54,56,57]. Nevertheless, it is crucial to note that the exact nature and extent of inflammation[25] in lipedema remain areas of active investigation, and the mechanistic underpinnings are not fully elucidated. In particular, it remains unclear whether the observed tissue inflammation is a cause or a consequence of lipedema.

**Statement 14:. "Numerous findings suggest that hormonal factors may contribute to the pathogenesis of lipedema".**

| | |
|---|---|
| Clarity of statement | 96,8% |
| **Level of agreement** | **96,9%** |
| Inclusion Rating | 92,1% |
| Strength of Evidence | 58,1% |
| Additional Evidence Needed | 56,3% |

Expert Panel Comment:

Hormonal factors are considered to contribute to the regulation of lipedema pathogenesis[58–60] as lipedema symptoms are reported to worsen during hormonal shifts[16]. Nevertheless, it is important to emphasize that precise pathophysiological mechanisms for conclusively linking hormonal factors to lipedema have not yet been fully elucidated[58,61,62].

**Statement 15:. "Several findings suggest that extracellular fluid volume might be elevated in lipedema-affected tissue compared to BMI-matched unaffected controls".**

| | |
|---|---|
| Clarity of statement | 87,1% |
| **Level of agreement** | **76,2%** |
| Inclusion Rating | 72,6% |
| Strength of Evidence | 46,8% |
| Additional Evidence Needed | 66,1% |

Expert Panel Comment:

Histological analysis[63], bioimpedance spectroscopy[64,65], in-vivo MRI imaging[27,66], and near-infrared fluorescent lymphatic imaging[67] indicate that lipedema might be characterized by an accumulation of excess extracellular fluid (ECF), possibly deriving from impaired capillaries or dysfunctional lymphatic vessels. Notably, clinical observation generally does not reveal visible edema in individuals with lipedema. A hypothesis has been postulated, suggesting that the absence of edema, despite an elevation in ECF, may be attributed to an increase in glycosaminoglycans and proteoglycans, which would bind the increased ECF[8].

Domain 3: Epidemiology

**Statement 16:. "Lipedema primarily affects biological females. Occurrence in biological males appears to be possible but rare".**

| | |
|---|---|
| Clarity of statement | 96,7% |
| **Level of agreement** | **88,7%** |
| Inclusion Rating | 88,9% |
| Strength of Evidence | 69,8% |
| Additional Evidence Needed | 33,9% |

Expert Panel Comment:

The prevailing understanding is that lipedema predominantly manifests in biological females. Limited case reports suggest the occurrence of lipedema in biological males, with cases frequently associated with hormonal abnormalities[68–72]. Additional research is warranted to explore the underlying factors contributing to this disparity.

**Statement 17:. "Hormonal changes may trigger or exacerbate the symptoms of lipedema".**

| | |
|---|---|
| Clarity of statement 1 | 00,0% |
| **Level of agreement** | **98,4%** |
| Inclusion Rating | 98,4% |
| Strength of Evidence | 63,9% |
| Additional Evidence Needed | 36,5% |

Expert Panel Comment:

Phases characterized by hormonal fluctuations, such as puberty, pregnancy, or menopause, are frequently temporally associated with the onset or exacerbation of lipedema symptoms[16,73]. The causality of hormonal changes and their potential to have an impact on lipedema symptoms[58,59], including the influence of other hormonal stimuli (such as hormone replacement therapy, contraceptives, etc.), remains uncertain[15]. Therefore, more comprehensive research, including longitudinal studies and investigations into the underlying biological mechanisms, is needed to establish a clearer understanding of the relationship between hormonal changes and lipedema.

**Statement 18:. "Lipedema is hereditary in some cases".**

| | |
|---|---|
| Clarity of statement | 96,8% |
| **Level of agreement** | **88,7%** |
| Inclusion Rating | 88,9% |
| Strength of Evidence | 63,9% |
| Additional Evidence Needed | 47,5% |

Expert Panel Comment:

A frequently observed positive family history, with prevalence ranging from 30% to 90%[13,42,73–76], indicates a hereditary nature of lipedema. An examination of familial clusters of lipedema patients has proposed an X-linked dominant inheritance pattern or, more likely, an autosomal dominant inheritance with sex restriction of a single dominant gene[77]. In addition, other studies have postulated an oligogenic inheritance model, suggesting the involvement of multiple gene variants associated with the phenotypic presentation of lipedema[78–84]. It is plausible that lipedema may manifest as either a primary type or as part of a syndromic hereditary condition[79].

**Statement 19:. "The prevalence of lipedema in the adult female population remains unknown. Estimates range from less than 1% to up to 12%".**

| | |
|---|---|
| Clarity of statement | 96,8% |
| **Level of agreement** | **79,0%** |
| Inclusion Rating | 83,9% |
| Strength of Evidence | 37,1% |
| Additional Evidence Needed | 66,1% |

Expert Panel Comment:

Accurately establishing the prevalence of lipedema presents a notable challenge owing to inconsistencies in diagnostic criteria, limited awareness among healthcare professionals and patients, variations in study populations (whether drawn from the general population or specialized institutions), geographical disparities, and differing methodological approaches. Current estimates within the adult female population underscore the considerable variability observed in reported figures[18,73,76,77,85–91].

Domain 4: Comorbidities & concomitant diseases

**Statement 20:. "Obesity is a frequently observed concomitant disease in patients with lipedema".**

| | |
|---|---|
| Clarity of statement | 98,4% |
| **Level of agreement** | **87,3%** |
| Inclusion Rating | 92,1% |
| Strength of Evidence | 62,9% |
| Additional Evidence Needed | 44,4% |

Expert Panel Comment:

Lipedema is often either misdiagnosed as or coexists with obesity, posing challenges in accurate differentiation. Research consistently

demonstrates a higher prevalence of obesity among individuals with lipedema compared to the general population, as assessed by BMI[13,40,77,92–94]. However, BMI is not an ideal tool to assess obesity in patients with lipedema due to the disproportionate body habitus inherent in lipedema[95], potentially contributing to divergent clinical evaluations and interpretations of existing literature. Nevertheless, when both conditions coexist, appropriate treatment must be provided for each as a separate disease. In addition, it is essential to recognize that the presence of obesity in lipedema patients may exacerbate symptoms and complicate management strategies. While the precise nature of the relationship between lipedema and obesity warrants further investigation, the recognition of this frequent concomitance is crucial for informing holistic and patient-centered approaches to care.

### Statement 21:. "Lipedema is not an obesity-related comorbidity".

| | |
|---|---|
| Clarity of statement | 98,4% |
| **Level of agreement** | **92,1%** |
| Inclusion Rating | 91,8% |
| Strength of Evidence | 69,4% |
| Additional Evidence Needed | 32,8% |

Expert Panel Comment:

Historically, lipedema was often mischaracterized as a simple consequence of obesity. Several studies have demonstrated that lipedema is distinct from obesity and should be regarded as a separate entity to facilitate accurate diagnosis and tailored therapeutic interventions[20,96–98]. Lipedema adipose tissue exhibits distinct morphological, molecular and metabolic characteristics compared to obesity-type adipose tissue[32,99]. In other words, lipedema is not an invariable comorbidity with obesity; rather, it may coincide with obesity[100].

### Statement 22:. "Body mass index (BMI) has limited value in distinguishing between lipedema and obesity. Therefore, it is advisable to utilize the Waist-to-Height Ratio (WHtR) to exclude or assess obesity".

| | |
|---|---|
| Clarity of statement | 95,2% |
| **Level of agreement** | **77,4%** |
| Inclusion Rating | 77,0% |
| Strength of Evidence | 64,5% |
| Additional Evidence Needed | 41,0% |

Expert Panel Comment:

The body mass index (BMI) for characterizing obesity is of limited value in lipedema patients, as it leads to falsely high values in the area of overweight or mild obesity due to the increase in adipose tissue in the extremities. Nevertheless, as in all areas of surgery, BMI may have a positive correlation with surgical complication rates. A more accurate assessment of disproportionate fat distribution and metabolic health can be achieved through the waist-to-height ratio (WHtR)[95,101–103]. However, it is essential to acknowledge that while WHtR may provide valuable insights, further research and consensus development are warranted to establish standardized criteria for its application in distinguishing lipedema from obesity across diverse populations[9,10].

### Statement 23:. "In cases where lipedema coincides with obesity, lipedema symptoms can be expected to persist after bariatric surgery".

| | |
|---|---|
| Clarity of statement | 91,9% |
| **Level of agreement** | **88,9%** |
| Inclusion Rating | 87,1% |
| Strength of Evidence | 58,7% |
| Additional Evidence Needed | 44,3% |

Expert Panel Comment:

Weight loss in patients with lipedema has been reported to reduce leg volume, improve quality of life and alleviate symptoms[92,104,105]. Conversely, current research indicates that despite significant weight loss achieved through bariatric surgery, symptoms related to lipedema often persist[92,100,106–108]. Studies suggest that the distinctive pathophysiology of lipedema may play a role in the ongoing persistence of these symptoms. In addition, the reduction in volume of the SAT in lipedema-affected extremities may be less pronounced compared to the trunk.

### Statement 24:. "Concomitant lymphostasis can develop in lipedema".

| | |
|---|---|
| Clarity of statement | 90,3% |
| **Level of agreement** | **88,7%** |
| Inclusion Rating | 86,9% |
| Strength of Evidence | 43,5% |
| Additional Evidence Needed | 52,5% |

Expert Panel Comment:

Reports suggest that lymphatic impairment leading to lymphostasis or lymphedema may co-occur at any stage of lipedema, further complicating the clinical presentation. The use of multiple modalities to assess lymphatic dysfunction in lipedema is ongoing, as well as the etiology of observed differences[27,66,67,109,110]. Without definitive evidence supporting a primary lymphatic dysfunction in lipedema, lymphostasis may be related to an imbalance in the production and removal of lymphatic fluid, causing a capacity overload (high-volume transport insufficiency, "Marsch thesis"[111]) as lymphoscintigraphic findings were not associated to the BMI[110,112]. Furthermore, lymphostasis may be related to coincident obesity or excessive adipose tissue expansion.

### Statement 25:. "Several findings suggest that the prevalence of hypothyroidism might be higher in lipedema patients than in non-lipedema populations with comparable BMI and age".

| | |
|---|---|
| Clarity of statement | 93,5% |
| **Level of agreement** | **73,3%** |
| Inclusion Rating | 71,0% |
| Strength of Evidence | 39,3% |
| Additional Evidence Needed | 71,0% |

Expert Panel Comment:

Recent publications highlighted a possible link between lipedema and hypothyroidism[13,14,19,90]. The demonstrated prevalence, ranging from 19% to 36%, surpasses that of comparable populations, adjusted for sex, age, and BMI (0.5–2.0%)[113]. However, the extent of a causal connection or whether hypothyroidism is merely an epiphenomenon of coincident obesity remains unclear[114].

### Statement 26:. "Lipedema might be associated with connective tissue disorders, such as hypermobility spectrum disorders".

| | |
|---|---|
| Clarity of statement | 96,8% |
| **Level of agreement** | **75,8%** |
| Inclusion Rating | 72,6% |
| Strength of Evidence | 37,7% |
| Additional Evidence Needed | 65,6% |

Expert Panel Comment:

Evidence suggests that women with lipedema exhibit decreased elasticity in both skin[115] and aorta[116], along with joint hypermobility[16,117,118] and muscle weakness[119]. There is speculation about a link between lipedema and connective tissue disorders,

specifically hypermobility spectrum disorders (HSD), although no conclusive underlying mechanism has been identified[120]. Considering the inconsistency of HSD in lipedema, a hypothesis has proposed categorizing it as a subtype ("*rusticanus Moncorps type*") of lipedema[115]. Further investigation is needed to confirm a definitive association between lipedema and HSD.

Domain 5: Impact on Quality of Life and Symptom Burden

**Statement 27:. "Lipedema can negatively impact mental health and overall quality of life".**

| | |
|---|---|
| Clarity of statement 1 | 00,0% |
| **Level of agreement** | **98,4%** |
| Inclusion Rating | 98,4% |
| Strength of Evidence | 70,5% |
| Additional Evidence Needed | 38,7% |

Expert Panel Comment:

The available literature consistently highlights the significant psychosocial burden associated with lipedema, emphasizing its adverse effects on mental health and overall quality of life[94,121–123]. The psychosocial implications of lipedema are multifaceted, encompassing impaired daily functioning, social withdrawal, and body image dissatisfaction[124,125]. Further evidence is required to elucidate how quality of life is affected across broader health domains and in diverse cultural contexts, as the psychosocial burden is exacerbated by insufficient support for affected individuals within many healthcare systems.

**Statement 28:. "If present, psychological involvement may be caused by lipedema-related symptoms rather than being the cause of those symptoms".**

| | |
|---|---|
| Clarity of statement | 93,4% |
| **Level of agreement** | **90,2%** |
| Inclusion Rating | 85,5% |
| Strength of Evidence | 48,4% |
| Additional Evidence Needed | 53,2% |

Expert Panel Comment:

The intricate interplay between psychological factors and the symptomatology of lipedema has garnered attention[123,126]. Existing literature suggests that the psychological burden experienced by individuals with lipedema may be a result of the challenges posed by the physical aspects and functional limitations associated with the disease. Conversely, psychological distress (e.g., after traumatic experiences) can potentially exacerbate symptomatology or influence coping mechanisms. Therefore, the relationship between psychological factors and lipedema is complex and warrants careful consideration.

**Statement 29:. "Missed or delayed diagnosis or management of lipedema negatively affects a patient's symptom burden, mental well-being, and overall quality of life".**

| | |
|---|---|
| Clarity of statement 1 | 00,0% |
| **Level of agreement 1** | **00,0%** |
| Inclusion Rating 1 | 00,0% |
| Strength of Evidence | 67,7% |
| Additional Evidence Needed | 31,1% |

Expert Panel Comment:

Lipedema may not consistently progress, yet certain factors, such as overall weight gain[42] and hormonal changes (e.g., during pregnancy), can contribute to long-term symptom exacerbation. A progression of the disease, marked by increased pain and disproportionate adipose tissue accumulation, along with unsuccessful therapeutic interventions and diets, heightens the risk of

developing depression and has a significant impact on the overall quality of life[15,94,121]. Therefore, it is advisable to pursue early diagnosis and implement timely treatment interventions[127].

**Statement 30:. "Missed or delayed diagnosis or management of lipedema increases the cost burden for patients and the healthcare system".**

| | |
|---|---|
| Clarity of statement 1 | 00,0% |
| **Level of agreement** | **93,3%** |
| Inclusion Rating | 91,8% |
| Strength of Evidence | 57,4% |
| Additional Evidence Needed | 49,2% |

Expert Panel Comment:

The economic impact of delayed diagnosis and suboptimal management in lipedema is a multifaceted concern, affecting both patients and the broader healthcare system. Research suggests that the complex nature and the fact that the condition is often overlooked contribute to delayed diagnosis, leading in turn to a protracted period of untreated symptoms[15]. This delay can result in increased healthcare utilization and costs due to the progression of the disease and the development of associated comorbidities[13]. Furthermore, the economic burden extends beyond direct medical costs, also encompassing expenses related to productivity loss, disability, and diminished quality of life[122].

Domain 6: Diagnostic approach

**Statement 31:. "The clinical diagnosis of lipedema relies on the patient's medical history, physical examination and exclusion of differential diagnoses".**

| | |
|---|---|
| Clarity of statement | 98,4% |
| **Level of agreement** | **98,4%** |
| Inclusion Rating | 98,4% |
| Strength of Evidence | 72,1% |
| Additional Evidence Needed | 35,0% |

Expert Panel Comment:

Accurate diagnosis of lipedema remains challenging due to its overlapping clinical features with other adipose tissue disorders and lymphatic conditions[128]. The reliance on comprehensive evaluation, including detailed medical history, thorough physical examination, and exclusion of differential diagnoses, underscores the complexity of diagnosing lipedema in clinical practice[4,8,9,46]. Notably, the absence of definitive diagnostic criteria and standardized assessment tools may contribute to delays in diagnosis and misidentification of the disease. Collaborative efforts are needed to establish consensus guidelines and improve diagnostic accuracy for the benefit of patients and healthcare providers.

**Statement 32:. "Currently, no imaging, serological or genetic tests, or clinical measurement instruments, are officially approved to verify the clinical diagnosis".**

| | |
|---|---|
| Clarity of statement | 98,4% |
| **Level of agreement** | **93,5%** |
| Inclusion Rating | 91,8% |
| Strength of Evidence | 66,1% |
| Additional Evidence Needed | 39,3% |

Expert Panel Comment:

Research efforts on different diagnostic modalities are underway to aid the clinical diagnosis of lipedema, including imaging, genetic, serologic, and bedside techniques[20,25,38,120,129–131]. While these investigations show promise in enhancing diagnostic precision and

differentiating from other conditions, it is crucial to acknowledge the current lack of official approval for any specific modality in verifying the clinical diagnosis of lipedema.

**Statement 33:. "Routine clinical exams should include standardized anthropometric measurements, such as waist-to-height ratio (WHtR), waist-to-hip ratio (WHR), and body mass index (BMI)".**

| | |
|---|---|
| Clarity of statement | 96,7% |
| **Level of agreement** | **82,3%** |
| Inclusion Rating | 75,8% |
| Strength of Evidence | 57,4% |
| Additional Evidence Needed | 43,3% |

Expert Panel Comment:

Although anthropometric measurements, including WHtR, WHR, and BMI, taken individually are not useful for diagnosing lipedema, taken together, they are essential in the assessment of overall health and are widely employed in routine clinical examinations[132]. Despite its limited significance in lipedema[133], BMI can support disease monitoring due to its simplicity in measurement and interpretation. While BMI does not reflect regional fat distribution, WHR is subject to age, sex, and ethnic variations. Conversely, WHtR has been identified as a valuable predictor of cardiometabolic risk, with studies suggesting its superiority over other indices, particularly in assessing central obesity[95,103,134,135].

**Statement 34:. "The clinical classification of lipedema into stages does not reflect the complete symptom severity".**

| | |
|---|---|
| Clarity of statement | 96,7% |
| **Level of agreement** | **95,1%** |
| Inclusion Rating | 93,4% |
| Strength of Evidence | 60,7% |
| Additional Evidence Needed | 52,5% |

Expert Panel Comment:

The current clinical stage classification of lipedema is based on the severity of palpation findings in the skin and subcutaneous tissue[136]. Palpable alterations correspond to a progressive enlargement of nodular and fibrotic tissue structures[39], accompanied by an increasing induration of the skin and subcutaneous tissue. While the current staging system primarily focuses on anatomical aspects, it does not adequately correlate with the nuanced and multifaceted nature of symptom severity in individuals with lipedema[137].

**Statement 35:. "The current clinical classification for lipedema into stages has limited relevance for the disease management".**

| | |
|---|---|
| Clarity of statement 1 | 00,0% |
| **Level of agreement** | **91,9%** |
| Inclusion Rating | 90,3% |
| Strength of Evidence | 54,1% |
| Additional Evidence Needed | 49,2% |

Expert Panel Comment:

The existing clinical classification of lipedema into stages does not reflect the individual differences in the total perceived discomfort due to lipedema, presenting challenges in guiding optimal disease management[9,10]. While stages have been suggested based on phenotypic characteristics and palpation findings[136], the practical utility of this classification in guiding therapeutic strategies is limited. To achieve a more nuanced and comprehensive approach to disease management, considerations should extend to factors such as pain intensity, quality of life, and functional impairment[98,137].

**Statement 36:. "A progression in the severity of lipedema-associated symptoms depends on various factors and is not universal".**

| | |
|---|---|
| Clarity of statement | 93,4% |
| **Level of agreement** | **95,1%** |
| Inclusion Rating | 95,1% |
| Strength of Evidence | 47,5% |
| Additional Evidence Needed | 57,4% |

Expert Panel Comment:

The trajectory of lipedema presents considerable heterogeneity, with varying degrees of symptom severity influenced by a complex interplay of factors. While some individuals may experience a progression in symptom severity over time, others may exhibit stable or fluctuating symptomatology. Existing literature suggests that factors such as hormonal changes, individual predisposition, overall weight gain[42] and lifestyle factors may contribute to the diversity in symptom progression[85]. However, it is essential to acknowledge the limited availability of long-term, prospective studies that comprehensively explore the multitude of factors influencing the natural history and disease course of lipedema.

**Statement 37:. "The excess limb volume in lipedema is generally not associated with obesity".**

| | |
|---|---|
| Clarity of statement | 93,2% |
| **Level of agreement** | **79,7%** |
| Inclusion Rating | 76,3% |
| Strength of Evidence | 39,0% |
| Additional Evidence Needed | 52,5% |

Expert Panel Comment:

The relationship between excess limb volume in lipedema and obesity is a complex and much-debated aspect within the current literature. While some authors suggest that limb volume increase in lipedema is not necessarily correlated with overall obesity[100], others propose a potential overlap between lipedema and obesity[9,10], indicating that the two conditions may coexist or influence each other. Given the high number of patients with lipedema who also suffer from concomitant obesity, excess limb volume can be attributed to both underlying conditions. The lack of a universally accepted definition for obesity in the context of lipedema and the varied methodologies used to assess limb volume and obesity contribute to the ongoing discourse. Further research and standardized criteria are warranted to elucidate the intricate interplay between limb volume excess in lipedema and obesity.

**Statement 38:. "The clinical classifications based on localization have only descriptive significance".**

| | |
|---|---|
| Clarity of statement | 95,1% |
| **Level of agreement** | **88,5%** |
| Inclusion Rating | 86,9% |
| Strength of Evidence | 39,3% |
| Additional Evidence Needed | 43,3% |

Expert Panel Comment:

While localization-based phenotype classifications (such as subtypes according to Meier-Vollrath[138] or form variant according to Herpertz[139]) provide a useful framework for characterizing the distribution of adipose tissue in lipedema[117,138,139], their clinical utility in predicting disease progression, response to treatments, or specific complications remains limited. Nevertheless, localization-based phenotype classifications serve as a descriptive tool facilitating the contextualization of clinical cases among experts, thus retaining their practical significance. Further research is needed to elucidate the practical implications and prognostic value of these classification

systems, taking into account the heterogeneity observed in patients with lipedema.

Domain 7: Treatment modalities

**Statement 39:. "All therapeutic interventions of lipedema aim at alleviating symptoms and preventing or delaying progression".**

| | |
|---|---|
| Clarity of statement | 95,1% |
| **Level of agreement** | **90,2%** |
| Inclusion Rating | 90,2% |
| Strength of Evidence | 55,7% |
| Additional Evidence Needed | 45,8% |

Expert Panel Comment:

Current therapeutic interventions for lipedema primarily focus on (1) symptom management and (2) disease progression prevention, rather than curative measures[9,46,97]. It is crucial to note that while existing treatment modalities for lipedema can provide meaningful symptom relief, they do not eliminate the underlying pathology of lipedema altogether.

**Statement 40:. "Comprehensive disease management requires a multidisciplinary approach tailored to individual needs, which may involve physicians, physical therapists, dietitians, and mental health professionals".**

| | |
|---|---|
| Clarity of statement | 96,7% |
| **Level of agreement** | **96,7%** |
| Inclusion Rating | 96,7% |
| Strength of Evidence | 63,9% |
| Additional Evidence Needed | 41,7% |

Expert Panel Comment:

The significance of a multidisciplinary approach in the comprehensive management of complex conditions, such as lipedema, is widely acknowledged in the literature[140]. Collaboration among different health care professionals is vital to address the diverse aspects of the disease, encompassing medical, rehabilitative, nutritional, and psychological dimensions. While there is a growing consensus on the importance of multidisciplinary care, further research is warranted to establish standardized protocols and assess the long-term effectiveness of such collaborative interventions in lipedema management.

**Statement 41:. "Lipedema pain and physical sensitivity in lipedema-affected areas have been reported to be reduced by bandaging, compression, complete physical decongestive therapy or other physical therapies (such as shock wave therapy), dietary changes, tailored exercise, and lipedema reduction surgery, with varying effect sizes and durations".**

| | |
|---|---|
| Clarity of statement | 86,7% |
| **Level of agreement** | **86,9%** |
| Inclusion Rating | 83,3% |
| Strength of Evidence | 45,9% |
| Additional Evidence Needed | 65,6% |

Expert Panel Comment:

Various interventions show potential for symptom relief in lipedema, but the quality of the supporting evidence is variable. The heterogeneity of study designs, patient populations and diagnostic criteria makes direct comparisons between different treatment modalities challenging. In addition, the duration of therapeutic effects remains a subject of investigation, with some interventions demonstrating sustained benefits over time, while others may necessitate ongoing management. It is important to emphasize that a universally applicable therapeutic regimen for lipedema has yet to be established. The choice of treatment modalities often depends on the individual patient's characteristics, including disease severity, comorbidities, and personal preferences[9].

**Statement 42:. "Conservative management of lipedema should include lifestyle and nutritional optimization, compression therapy, and exercise to alleviate symptoms and improve quality of life".**

| | |
|---|---|
| Clarity of statement | 95,1% |
| **Level of agreement** | **93,4%** |
| Inclusion Rating | 90,0% |
| Strength of Evidence | 63,3% |
| Additional Evidence Needed | 44,1% |

Expert Panel Comment:

Lifestyle and nutritional changes play a pivotal role, aiming to address potential exacerbating factors and enhance overall well-being[104,141,142]. Compression therapy has demonstrated effectiveness in improving symptoms, contributing to enhanced patient comfort and mobility[143-145]. Tailored exercise regimes, such as aquatic exercise, are often recommended to promote lymphatic flow[146] and muscle strength without placing excessive strain on affected limbs[145,147]. While individual responses to these interventions may vary, a comprehensive strategy incorporating these elements is a cornerstone in the holistic care of lipedema patients.

**Statement 43:. "Active self-management can help control lipedema-related symptoms and improve the overall quality of life".**

| | |
|---|---|
| Clarity of statement | 95,1% |
| **Level of agreement** | **91,7%** |
| Inclusion Rating | 90,0% |
| Strength of Evidence | 36,1% |
| Additional Evidence Needed | 59,3% |

Expert Panel Comment:

Existing literature suggests that active self-management practices play a crucial role in mitigating lipedema-related symptoms and enhancing the overall quality of life for affected individuals[9]. Patient education and training are critical components of effective self-management. However, it is important to note that the evidence base for specific self-management strategies is still evolving, and further research is needed to establish comprehensive guidelines for effective self-management in lipedema.

**Statement 44:. "Although Complex (also known as Complete) Decongestive Therapy (CDT) can be an important and effective treatment even for early-stage lipedema, not all components are required for every patient".**

| | |
|---|---|
| Clarity of statement | 95,1% |
| **Level of agreement** | **86,9%** |
| Inclusion Rating | 86,9% |
| Strength of Evidence | 39,3% |
| Additional Evidence Needed | 53,3% |

Expert Panel Comment:

The key components of Complex (also known as Complete) Decongestive Therapy (CDT) include (1) compression therapy, (2) manual lymphatic drainage (MLD), (3) decongestive exercise, (4) skin care, and (5) empowerment/self-management. Compression therapy for lipedema aims to reduce pain and alleviate subjective symptoms, especially when combined with physical activity, demonstrating positive effects[145,148]. There is no evidence supporting the sole use of MLD for

therapeutic benefits in lipedema, as it has only been studied in various combination forms with other therapies[143,144,149]. However, it has been shown to have a sympatholytic effect, increase pain tolerance, and elevate pain thresholds[9]. Not all lipedema patients may require every component of CDT, and tailoring therapeutic interventions based on the specific needs and characteristics of each patient is crucial.

**Statement 45:. "Nutritional guidance can help patients manage their weight, optimize overall health, reduce lipedema-associated symptoms and improve their response to therapeutic interventions".**

| | |
|---|---|
| Clarity of statement | 96,7% |
| **Level of agreement** | **93,4%** |
| Inclusion Rating | 93,4% |
| Strength of Evidence | 65,6% |
| Additional Evidence Needed | 47,5% |

Expert Panel Comment:

The existing literature suggests that addressing chronic inflammation in lipedema, especially with concomitant obesity, through patient education on pro-inflammatory triggers and recommending an anti-inflammatory Mediterranean or ketogenic diet[141,150–158]. Elevated insulin levels in both conditions can contribute to lipogenesis and inflammation, emphasizing the importance of a diet that avoids blood glucose and insulin spikes[159–161]. Limited studies exist on lipedema-specific diets, but reviews highlight the potential benefits of ketogenic diets, including weight reduction, decreased adipose tissue, and symptom relief[105,142,162–166]. The absence of pro-inflammatory blood glucose spikes in ketogenic diets is proposed to make them more effective in combating lipedema inflammation[151].

**Statement 46:. "Although pathological subcutaneous adipose tissue in lipedema is known to be largely resistant to dietary interventions, addressing overall weight loss in coincident obesity may result in symptom improvement".**

| | |
|---|---|
| Clarity of statement | 95,0% |
| **Level of agreement** | **93,3%** |
| Inclusion Rating | 93,2% |
| Strength of Evidence | 62,1% |
| Additional Evidence Needed | 40,7% |

Expert Panel Comment:

Although lipedema SAT is considered largely diet-resistant[19], improvements in lipedema-associated symptoms have been demonstrated through a combination of dietary modification, weight reduction and exercise[141]. Conversely, it has been reported that in patients with coincident obesity, bariatric surgery–induced weight loss and reductions in BMI did not result in significant improvement of lipedema-associated symptoms in the absence of dietary modification[100]. To promote optimal health and mobility, it is advisable to provide nutritional guidance and, when applicable, engage in obesity therapy aligned with clinical guidelines[97].

**Statement 47:. "Considering that obesity worsens the manifestations of lipedema, disease management should include weight optimization, with a focus on waist-to-height ratio (WHtR) and waist-to-hip ratio (WHR)".**

| | |
|---|---|
| Clarity of statement | 91,8% |
| **Level of agreement** | **83,3%** |
| Inclusion Rating | 80,3% |
| Strength of Evidence | 50,8% |
| Additional Evidence Needed | 51,7% |

Expert Panel Comment:

Maintaining a normal weight can help prevent a progression of lipedema-associated symptoms, preserve mobility, and reduce the risk of developing osteoarthritis[97]. Obesity classification typically relies on BMI[132]. However, in the context of lipedema and in line with statement 33, utilizing alternative anthropometric measures—such as WHtR—may be more appropriate for assessing or excluding overweight and obesity, particularly in the context of metabolic health categorization[95,167].

**Statement 48:. "Psychological and social support, addressing body image issues, mental well-being, and coping strategies, can be important to address the symptom burden of patients living with lipedema".**

| | |
|---|---|
| Clarity of statement | 96,7% |
| **Level of agreement** | **98,3%** |
| Inclusion Rating | 96,7% |
| Strength of Evidence | 68,3% |
| Additional Evidence Needed | 37,3% |

Expert Panel Comment:

Acknowledging the profound impact of lipedema on patients, addressing psychological and social dimensions is vital for comprehensive care. Studies emphasize the multifaceted psychosocial challenges, including impaired daily functioning, social withdrawal, and mental health implications[122,123,168]. Interventions targeting these aspects have demonstrated efficacy in improving overall well-being and coping mechanisms[121,169].

**Statement 49:. "Tailored exercises, such as physical activity in water, walking, and yoga, can help maintain mobility, address lipedema related symptoms and support weight management in individuals with lipedema".**

| | |
|---|---|
| Clarity of statement | 95,0% |
| **Level of agreement** | **91,7%** |
| Inclusion Rating | 90,2% |
| Strength of Evidence | 56,7% |
| Additional Evidence Needed | 42,6% |

Expert Panel Comment:

A growing body of literature underscores the potential benefits of exercise interventions for individuals with lipedema, emphasizing improvements in physical function, mobility, symptom management and quality of life[143,144,149,170–172]. Specifically, water-based activities have shown promise in reducing pain and enhancing mobility, possibly due to effects and physical properties of water, such as hydrostatic pressure and buoyancy, and consequently reduced impact on joints[147,173,174]. Moreover, structured exercise—such as walking and yoga—have been associated with improved muscular strength, and psychological well-being, factors that are particularly relevant for individuals with lipedema[119,175,176].

**Statement 50:. "Currently, there is no evidence for the effectiveness of any pharmacological interventions in treating lipedema".**

| | |
|---|---|
| Clarity of statement 1 | 00,0% |
| **Level of agreement** | **93,3%** |
| Inclusion Rating | 93,2% |
| Strength of Evidence | 58,3% |
| Additional Evidence Needed | 44,1% |

Expert Panel Comment:

There is no evidence to support the efficacy of pharmaceutical interventions in alleviating lipedema symptoms. Supplements with potential antioxidative, edema-reducing, or immunomodulatory effects

have been explored for managing lipedema symptoms. However, the existing evidence supporting their efficacy is limited[151,153,155,156,177–179].

**Statement 51:. "In cases where lipedema coincides with obesity and metabolic disease, it is advisable to prioritize treatment for obesity before considering lipedema reduction surgery".**

| | |
|---|---|
| Clarity of statement 1 | 00,0% |
| **Level of agreement** | **93,3%** |
| Inclusion Rating | 91,7% |
| Strength of Evidence | 66,7% |
| Additional Evidence Needed | 40,7% |

Expert Panel Comment:

The relationship between obesity, metabolic disease, and lipedema is complex, necessitating a nuanced approach to treatment decisions. Existing literature highlights the bidirectional interplay between obesity and lipedema, with obesity potentially exacerbating lipedema symptoms[180] and vice versa due to impaired mobility, pain or psychological burden linked to lipedema[13,40,172,174,181]. While addressing obesity is crucial for overall health, it is essential to recognize that lipedema is a distinct condition with unique pathophysiological features.

The sequential or simultaneous management of both conditions may be warranted, acknowledging that successful outcomes often require a multidisciplinary approach[9]. When considering lipedema reduction surgery, it is generally advisable to address concurrent obesity first[182], as a lower BMI correlates with improved outcomes[74] and may mitigate perioperative risks[183,184]. However, in some cases, it may not be universally applicable to postpone lipedema reduction surgery until obesity or metabolic health has been addressed, as the severity of the condition, the patient's goals and general health may vary from case to case[100,185].

**Statement 52:. "Lymph vessel-sparing lipedema reduction surgery should be considered when there is potential for a positive impact on lipedema-related symptoms".**

| | |
|---|---|
| Clarity of statement | 86,7% |
| **Level of agreement** | **89,7%** |
| Inclusion Rating | 87,9% |
| Strength of Evidence | 58,6% |
| Additional Evidence Needed | 49,2% |

Expert Panel Comment:

Although conservative treatment can alleviate lipedema symptoms, it does not achieve long-lasting benefits and cannot prevent the progression of the disease. Lipedema reduction surgery, or liposuction, is currently the only technique for removing abnormal lipedema tissues and slowing a potential progression of the disease. Recent uncontrolled before-and-after studies reported positive results in reducing extremity size and alleviating lipedema-associated symptoms such as spontaneous pain, easy bruising, sensitivity to pressure, impairment in quality of life, restrictions to mobility, edema, sensation of tightness, and overall improvement in patients' quality of life[186–189]. However, currently, there is a lack of randomized controlled trials and long-term follow-up studies assessing the clinical effectiveness and safety of lipedema reduction surgery compared to no treatment or alternative approaches in managing lipedema. Ongoing studies have not yet yielded definitive results[190,191].

**Statement 53:. "Surgical interventions should be performed by healthcare providers with extensive knowledge in lipedema management, including conservative treatments, as part of an integral approach".**

| | |
|---|---|
| Clarity of statement | 98,3% |
| **Level of agreement** | **98,3%** |
| Inclusion Rating | 98,3% |
| Strength of Evidence | 53,3% |
| Additional Evidence Needed | 43,1% |

Expert Panel Comment:

In contrast to aesthetic liposuctions, the primary goal of lipedema reduction surgery is a medically indicated, functional intervention for symptom reduction through subtotal resection of the presumed pathological, subcutaneous fat tissue in the limbs affected by lipedema. Therefore, lipedema reduction surgery, integrated with conservative management, should be performed according to clinical best practice outlined by medical associations and developed specifically for lipedema[4–6,8–10].

Domain 8: Future directions

**Statement 54:. "Raising awareness about lipedema within the medical community and wider society is essential to reduce misdiagnosis and stigma".**

| | |
|---|---|
| Clarity of statement 1 | 00,0% |
| **Level of agreement** | **98,3%** |
| Inclusion Rating | 96,7% |
| Strength of Evidence | 66,7% |
| Additional Evidence Needed | 30,0% |

Expert Panel Comment:

Raising awareness about lipedema is critical for timely diagnosis and management, as patients often experience prolonged diagnostic delays and encounter misconceptions surrounding the condition[13,15]. Adequate education and awareness programs targeted at healthcare professionals can enhance recognition and understanding of lipedema's distinct features, reducing the likelihood of misdiagnosis. Furthermore, public awareness initiatives play a crucial role in diminishing the stigma associated with lipedema, fostering empathy, and promoting a supportive environment for affected individuals[169].

**Statement 55:. "A comprehensive standardized case report form (CRF) should be developed to improve consistency in diagnosis, lipedema reporting, follow-up, and research, and to facilitate, for example, both cohort and longitudinal studies".**

| | |
|---|---|
| Clarity of statement 1 | 00,0% |
| **Level of agreement** | **96,7%** |
| Inclusion Rating | 95,0% |
| Strength of Evidence | 55,9% |
| Additional Evidence Needed | 42,4% |

Expert Panel Comment:

Standardized data collection tools are crucial for advancing our understanding of lipedema and fostering collaboration across diverse clinical and research domains. Implementation of a CRF can potentially address existing variations in diagnostic criteria, improve accuracy in reporting, and contribute to the creation of a robust foundation for clinical studies or registries.

**Statement 56:. "Further research is needed to elucidate the biological mechanisms underlying lipedema, leading to objective diagnostic criteria and targeted therapies".**

| | |
|---|---|
| Clarity of statement | 98,3% |
| **Level of agreement 1** | **00,0%** |
| Inclusion Rating | 98,3% |
| Strength of Evidence | 57,6% |
| Additional Evidence Needed | 44,8% |

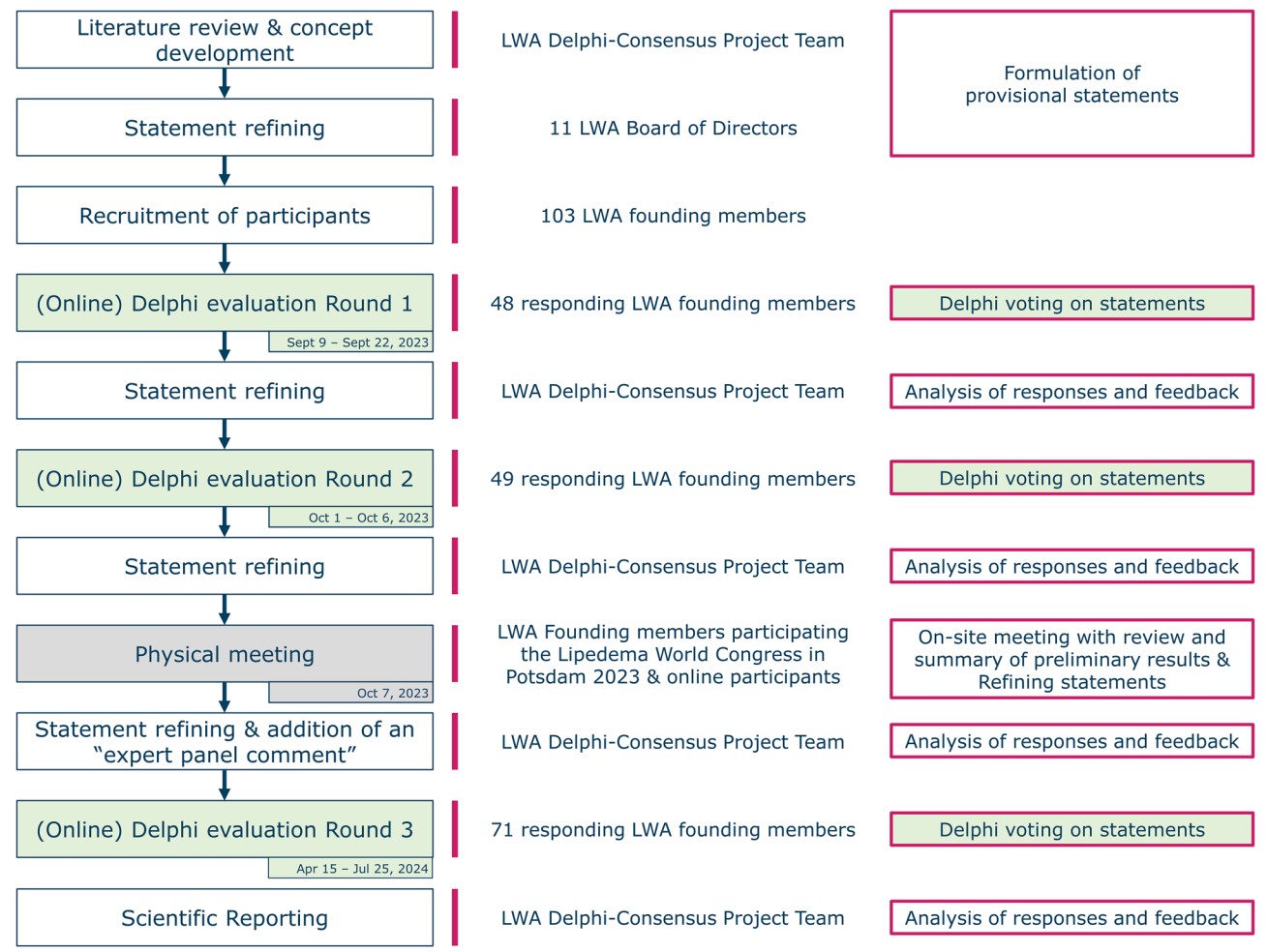

**Fig. 1 |** The modified Delphi method used in this study is shown schematically.

**Expert Panel Comment:**

The call for continued research into the biological mechanisms of lipedema is crucial for advancing our understanding of this complex disease. A comprehensive understanding of these mechanisms is essential for the development of objective diagnostic criteria, robust diagnostic markers and targeted therapeutic interventions. Collaborative efforts between clinicians, researchers, and advocacy groups are integral to driving progress in this field and improving outcomes for individuals affected by lipedema.

**Statement 57:. "Studies are required to validate diagnostic modalities for lipedema, which assess their reproducibility, sensitivity, and specificity".**

| | |
|---|---|
| Clarity of statement 1 | 00,0% |
| **Level of agreement 1** | **00,0%** |
| Inclusion Rating | 96,7% |
| Strength of Evidence | 58,6% |
| Additional Evidence Needed | 41,4% |

Expert Panel Comment:

While the need for rigorous validation of diagnostic modalities for lipedema is evident, the current landscape lacks comprehensive studies assessing the reproducibility, sensitivity, and specificity of existing diagnostic tools. The urgency for such research is underscored by the diverse clinical presentations and challenges in accurately identifying lipedema, often leading to delayed or incorrect diagnoses. To enhance diagnostic accuracy, future studies should aim to establish standardized protocols for evaluating the reliability and consistency of diagnostic modalities. These protocols should consider the variability in disease manifestations and demographic factors, ensuring applicability across diverse patient populations. Collaborative efforts, such as multicenter trials, could facilitate the accumulation of robust evidence, fostering a more nuanced understanding of diagnostic performance.

**Statement 58:. "Long-term studies are required to assess the efficacy and safety of treatment modalities for lipedema".**

| | |
|---|---|
| Clarity of statement 1 | 00,0% |
| **Level of agreement 1** | **00,0%** |
| Inclusion Rating | 98,3% |
| Strength of Evidence | 57,9% |
| Additional Evidence Needed | 46,4% |

Expert Panel Comment:

While some studies have provided insights into short-term outcomes, a comprehensive understanding of the long-term effects, durability of response, and potential adverse events is crucial for guiding evidence-based clinical decision-making. The call for long-term studies to assess treatment modalities in lipedema is substantiated by the current gaps in knowledge. Future research endeavors should prioritize extended follow-up periods to inform clinicians, researchers, patients, and service providers about the enduring impact and safety considerations associated with various therapeutic approaches.

**Statement 59:**. "**Collaborative efforts between patients, researchers, clinicians and advocacy groups are crucial for advancing knowledge. Translational practice-based application of research knowledge should improve patient care**".

| | |
|---|---|
| Clarity of statement 1 | 00,0% |
| **Level of agreement** | **98,3%** |
| Inclusion Rating | 95,0% |
| Strength of Evidence | 56,9% |
| Additional Evidence Needed | 36,2% |

Expert Panel Comment:

Collaborative engagement among patients, researchers, clinicians, and advocacy groups is pivotal for advancing our understanding of complex conditions like lipedema. Such an inclusive approach would foster a more comprehensive perspective, incorporating diverse insights and experiences, which is essential for addressing the multifaceted nature of the disease. Moreover, patient and public involvement in the research process enhances the relevance and applicability of findings to real-world clinical scenarios[192–194].

The translation of research knowledge into practice is a cornerstone for improving patient care. By bridging the gap between scientific discoveries and clinical applications, we can expedite the integration of evidence-based interventions into routine care, ultimately benefiting individuals affected by lipedema. Emphasizing translational practice ensures that the latest advancements directly influence patient outcomes and treatment strategies, facilitating a more responsive and patient-centered healthcare approach.

## Discussion

This is the first Delphi study undertaken to systematically evaluate and report the level of agreement on best-practice recommendations for patients with lipedema. The study represents a foundational milestone in developing consensus among international experts about lipedema disease characteristics, diagnostic approaches, and treatment modalities. This marks a significant initial step toward enhancing clarity and guiding future policy development and research on lipedema.

The international approach also allowed for a range of perspectives to be considered, so that patients benefit from shared experiences that improve clinical care in different health care systems. Unlike other consensus documents or best-practice guidelines, these statements, pertaining to the definition and management of lipedema were developed through a transparent process that involved a broad panel of international experts and representatives from patient organizations (Fig. 1). The use of the Delphi methodology enhances the strength and credibility of these statements, as it is a validated tool for developing best-practice guidance based on collective expert opinion when research is limited[195].

Considering the level of available evidence and the rapid pace of innovation in the field, the authors anticipate a limited lifespan for the statements presented in their current form. Therefore, the LWA is committed to maintaining this document as a "living document" that will require periodic review, expansion, and potential revision as new evidence emerges. In addition, although experts from 19 countries participated, significant populations affected by lipedema—such as those in Asia and Africa—remain underrepresented in this work. It is the authors hope that the global community grows in awareness of lipedema and future participation in best-practice guidelines.

At this current time, this document also aims to advance lipedema-related research by (1) underscoring the need for clear diagnostic criteria, (2) emphasizing the importance of standardized reporting, and (3) serving as a resource to engage large, traditional health research funding bodies in recognizing lipedema as a women's health disparity that requires attention. Considering the limited evidence in several areas, a potential next step to enhance the quality of evidence could involve identifying priority research areas, based on findings from the Delphi study with expert focus groups in relevant fields. However, the authors hope that the current consensus will provide an evolving framework to guide future clinical and research advances in the field. In addition, standardized reporting was identified as essential for improving data comparability and reproducibility. To this end, a case report form (CRF) could be developed specifically for use in lipedema research studies.

Through three voting rounds, consensus was reached on 59 statements regarding the definition and management of lipedema. Overall, a high level of agreement was achieved for the majority of aspects, with 36 statements achieving 90–100% consensus, 17 statements achieving 80–90% consensus, and 6 statements achieving 70% consensus (Table 1). However, it became evident that many of the statements were supported by a low level of evidence, highlighting the urgent need for high-quality research on lipedema.

To conduct such research, the diagnostic criteria or disease definition must be clear and standardized across research groups. During the on-site meeting in Potsdam and through feedback collected from the anonymous Delphi rounds, significant discrepancies in assessments were identified in certain aspects of the definition and pathophysiology of lipedema.

A key point of contention is the characterization of "pain" as a cardinal symptom of lipedema. A better understanding of the distinctly altered sensory sensitivity in patients with lipedema may help resolve any existing definitional differences and contribute to a more standardized definition of lipedema. Clarity on the experience of pain in individuals with lipedema would inform more effective treatment approaches.

Another aspect that generated significant variability in assessments and was the focus of extensive discussion was the possible presence of (protein-bound) extracellular fluid in lipedema, the eponymous "edema" in lipedema. Among all the statements, in the final round of the Delphi study, this was one of the top areas where participants identified a greater need for further research to enable a definitive conclusion. The authors believe that obtaining higher-quality evidence about edema in lipedema would be instrumental in standardizing diagnostic criteria across different geographic regions and disciplines, ultimately leading to a more unified understanding of the disease "lipedema".

This document has several strengths. The founding members of the LWA are recognized experts in the management of lipedema, and participants in the Delphi survey have extensive experience (Table 3). The diverse professional background of the participants—including conservative and surgical healthcare providers, researchers, and patient representatives—highlights the significant applicability of this project (Table 4). Furthermore, the geographical representation of participants from 19 countries across five continents (Table 2) reinforces the document's assertion of universal applicability. The anonymized methodology aimed to reduce the potential for introducing bias and enhance the validity of the consensus process. However, to preserve anonymity throughout the Delphi process, participant interactions were restricted, which limited opportunities for discussion and refinement of statements, aside from the feedback that was available to the project team.

In terms of limitations of the present study, it is of critical importance to note that the statements made are predominantly based on limited evidence. Additional constraints arise from the methodology employed. By definition, the use of a modified Delphi approach and reliance on expert opinions may introduce potential biases in the interpretation of the literature review, particularly since a substantial number of available publications were authored by participants in the Delphi process. Furthermore, the response rates in the initial two rounds of the Delphi survey—48 and 49 out of 103 LWA founding members—although balanced in terms of geographical distribution

and professional background, limit the representativeness of the findings (Tables 2–4). The in-person meeting after Round 2 during the dedicated session at the Lipedema World Congress in Potsdam on October 7, 2023, significantly increased the project's visibility and attention. This led to a higher number of participants in the third Delphi round, ensuring the final results are well-founded and reliable.

## Methods

The study was conducted in accordance with the principles of Good Clinical Practice and the Declaration of Helsinki. Ethics committee approval was not required, as no individual health data were used. Study procedures and reporting followed the DELPHISTAR reporting guidelines for Delphi studies in health research[196]. A completed DEL-PHISTAR checklist is provided in the Supplementary Information. No further consulting in regard to the method took place.

### Project team selection

This project was led by the Lipedema World Alliance (LWA), a non-for-profit organization composed of healthcare professionals, researchers, and patient association representatives dedicated to addressing lipedema globally. Experts invited by the LWA Board of Directors to join the Delphi-Consensus Project Team were selected based on their expertise and scholarly contributions in the field. Special attention was given to ensuring a diverse perspective of participants, considering demographic factors (age, sex, country), specialty, and level of experience.

### Participant selection

All 103 LWA founding members were invited to participate in the Delphi process via email. Founding Members included (1) physicians/medical doctors, healthcare professionals, and researchers dedicated to lipedema and related pathologies, as well as (2) national and regional organizations registered as not-for-profit and dedicated to fighting lipedema and/or related pathologies, comprising patients and/or family members and/or caregivers and/or professionals. The geographical distribution of the founding members reflects the project's wide-reaching, transnational, and intercontinental scope.

### Modified Delphi process

Delphi statements were developed by the project team around 8 domains regarding lipedema: (1) definition and leading symptoms, (2) pathophysiology, (3) epidemiology, (4) comorbidities and concomitant diseases, (5) impact on quality of life and symptom burden, (6) diagnostic approach, (7) treatment modalities, and (8) future directions.

A three-round Delphi process design was used (Fig. 1). Rounds 1 and 2 were survey-based via an online platform (SurveyMonkey Inc., San Mateo, CA, USA). Round 3 involved an on-site meeting held as part of the *Lipedema World Congress* on October 7th 2023 in Potsdam, Germany and a final follow-up survey. Although the administrator (PK) was aware of the identities of all respondents for tracking purposes, no identifying information was shared with the Delphi-Consensus Project Team or LWA Board members. For the online voting, all participants remained anonymous to each other throughout the Delphi study. Each stage of the Delphi process was systematically reported to and approved by the LWA Board of Directors.

During each Delphi round, statements were presented to the participant via an online survey. For each statement, participants evaluated 5 criteria: (i) the clarity of the statement, (ii) the degree of agreement with the statement, (iii) the recommendation for inclusion in the final document, (iv) the level of evidence supporting the statement, and (v) whether additional evidence is required to make a conclusive evaluation of the statement. Statement clarity and the need for additional evidence were assessed by yes/no response. Agreement with each statement and recommendation for inclusion were evaluated using a five-point Likert scale, as follows: (1) strongly disagree,

(2) disagree, (3) neither agree nor disagree, (4) agree, and (5) strongly agree. Evaluation of the strength of evidence for each statement was similarly conducted using a five-point Likert scale: (1) very low quality, (2) low quality, (3) neither high nor low quality, (4) high quality, and (5) very high quality. The degree of agreement on each criterion was calculated as the percentage of votes with a score of ≥ 4 on the five-point Likert scale or by the percentage of yes responses.

Consensus, or sufficient agreement, for a given statement was defined a priori as greater than 70% of expert agreement on criterion (ii), along with more than 70% of participants recommending inclusion in the document on criterion (iii)[197–200]. Criteria (i), (iv), and (v) are not relevant to the consensus threshold but serve to contextualize each statement. Statements that did not meet the established consensus threshold in the subsequent Delphi rounds were excluded from the final document but were included in the supplementary material to reflect the expert's preferences[199,201]. In each round of voting, participants were also encouraged to provide comments to explain their respective voting scores, and responses were anonymized. The LWA Delphi-Consensus Project Team reviewed the voting results and comments after each round, refining the statements for subsequent rounds of voting. If the comments implied that language-related misunderstandings contributed to insufficient agreement or that specific phrasing could be modified to achieve a higher level of consensus, the respective statement was adjusted accordingly and re-submitted for voting in the subsequent Delphi round (round 1: #17, #21, #23, #24, #27, #29, #30, #33, #35, #40, #41, #42, #54, #55, #56; round 2: #15, #24, #25, #35, #38).

Delphi evaluation round 1 consisted of 62 statements. All statements were developed by the LWA Delphi-Consensus Project Team and reviewed by the entire LWA board of directors. The online survey evaluation on the statements were distributed between September 9th and September 22nd 2023.

Round 2 included the aggregate results from round 1 and adjusted a total of 29 statements (#5, #7, #10, #13, #14, #15, #19, #20, #21, #24, #25, #26, #28, #32, #33, #34, #35, #36, #37, #39, #41, #43, #44, #47, #50, #51, #52, #57, #59) according to the comments from round 1. Five statements were removed from further evaluation due to an insufficient level of agreement and/or recommendation for inclusion in the final document. Two additional statements (#48, #62) were incorporated based on feedback received, resulting in a total of 59 statements. Online distribution took place between October 1st 2023 and October 6th 2023. Participants were provided with a summary of comments when necessary to understand changes from the previous version.

After the second Delphi evaluation round all LWA founding members were invited to participate in an on-site panel discussion during a dedicated session at the Lipedema World Congress in Potsdam on October 7th 2023. Online participation was also facilitated. During the on-site meeting, results showing the percentage of votes scoring ≥ 4 on the five-point Likert scale or the percentage of yes responses per statement were shared and related comments were reviewed. Statements that reached consensus in the previous e-Delphi rounds were endorsed with minimal discussion. Statements that did not reach consensus in the previous e-Delphi rounds were discussed, and, when appropriate, reformulated or removed altogether from further evaluation rounds.

Finally, to enhance clarity for the final round (Round 3) of the survey, input from participants of the in-person meeting, along with open-ended feedback from previous Delphi rounds, were both utilized to draft corresponding "expert panel comments" for each of the remaining statements. In addition, 26 statements were revised based on a combination of the discussions held in Potsdam and comments from the online surveys (Statements #2, #3, #4, #6, #10, #16, #18, #19, #24, #31, #33, #36, #39, #40, #41, #42, #44, #46, #47, #49, #51, #52, #53, #54, #55, #57). All remaining 59 statements were included in the final survey after appropriate revision. The online distribution of the survey occurred between April 15th 2024, and July 25th 2024. After the

third and final round of the Delphi process, no further modifications were made to the statements, apart from any grammatical changes to implement people-first language. Revisions following careful consideration of all feedback were limited exclusively to the "expert panel comments".

### Reporting summary

Further information on research design is available in the Nature Portfolio Reporting Summary linked to this article.

## Data availability

The data that support the findings of this study are available from the corresponding author upon request without restrictions.

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

## Acknowledgements

We thank the participants involved in this Delphi study. All participants were not paid an honorarium for their contributions to the Delphi study or any ensuing publication activities. The LWA Delphi-Consensus Project Team, assigned by the LWA Board members, consisting of the authors of this paper, takes full responsibility for the contents of this publication. In addition to the author team, the below specialists participated in the voting process of the Delphi methodology, but not in interpretation of the results, and agreed to be mentioned here:Amato, Alexandre; Belgrado, Jean-Paul; Boccardo, Francesco; Brenner, Erich; Burgos de la Obra, Enrique; Cannataro, Roberto; Cestari, Marina; Dimakakos, Evangelos; Drehmer Rieger, Eraci; Dudek, Joanna; Fetzer, Sharie; Forster, Kathryn Jennifer; Giordano, Valeria; Gupta, Puneet; Helmbrecht, Susanne; Hucho, Tim; Kahn, Linda Anne; Keith, Leslyn; Kollecker, Karsten; Lourenço Marques, Manuela; Majewski, Heike; Mazzolai, Lucia; Michelini, Serena; Ott, Johannes; Rapprich, Stefan; Reina Gutierrez, Lourdes; Schrader, Klaus; Schulz, Tim; Seo, Catherine; Szuba, Andrzej; Tofteng Hansen, Lene; Ure, Christian; Warrilow, Mary; Wigg, Jane; Wright, Thomas. We are also grateful to the additional specialists who participated in the Delphi study and chose to remain anonymous. Special thanks are extended to the representatives of patient advocacy organizations, whose invaluable contributions were instrumental in the successful execution of this project. Although no patient representatives were directly involved in the scientific evaluation or manuscript preparation, their feedback greatly enhanced many of the formulated statements. The following patient advocacy organizations participated in the voting process of the Delphi methodology but did not contribute to the interpretation of the results: Brazil: Associação Brasileira de Lipedema (ABRALI). Denmark: Dansk Lymfødem Forening (DALYFO). Germany: Lymphselbsthilfe e.V. Italy: LIO Lipedema Italia APS/ETS – Associazione Italiana Lipedema. Norway: Norsk lymfødem og lipødemforbund (NLLF). Portugal: Associação Nacional de Doentes Linfáticos (andLINFA). United Kingdom: Lipoedema UK. United Kingdom: Talk Lipoedema. USA: Fat Disorders Resource Society. USA: The Lipedema Project. We would also like to extend our gratitude to all healthcare professionals, and researchers affiliated with the LWA who did not participate in the voting

process of the Delphi methodology, but have expressed their support and endorsement for this document: Bertelli, Matteo; Bruno, Agostino; Haag, Margareta; Henriques, Margarida; Puigdellivol, Cristina; Rajewski, Diana; Torres Gonçalves, Daniel. Furthermore, we extend our sincere appreciation to organizations affiliated with the LWA who were not directly involved in the Delphi Project but have expressed their support and endorsement for this document: France: Association de la maladie du Lipœdeme France (AMLF). Germany: LY.SEARCH. USA: American Lipedema Association. USA: Lipedema Foundation.

## Author contributions

S.M. and P.K. developed the concept and designed the survey. P.K. collected and analyzed the data and reviewed all comments submitted as part of the three survey rounds. All authors discussed the results and implications of the respective Delphi rounds and commented on statement refinements. P.K. drafted the manuscript. R.C., G.F., I.F.C., M.C., R.S., T.K., J.L.S., K.L.H., M.G., and S.M. edited and commented on the manuscript at all stages. All authors contributed to the full Delphi study and contributed to significant amendments to the final manuscript.

## Competing interests

Karen L. Herbst discloses her role as an independent contractor for Lympha Press. The remaining authors declare no competing interests.
