## [Transparent Peer Review file · Nature Communications]

Lipedema World Alliance Delphi Consensus-Based Position Paper on the Definition and Management of Lipedema: Results from the 2023 Lipedema World Congress in Potsdam

Corresponding Author: Dr Philipp Kruppa

Version 0:

Reviewer comments:

Reviewer #1

(Remarks to the Author)

The Delphi Consensus-Based Position Paper shows noteworthy consistent results and is a significant review of statements regarding the definition and management of lipodema, building on the established literature. A comprehensive literature search justified the statements made, and the consensus highlights the need for further research in this condition.

The process takes an inclusive approach and focuses on the diversity component. However, the data still lacks gaps to represent the spectrum of protected characteristics in the expert panel (such as gender and age).

There were no obvious flaws in the data analysis, but some of the expert panel recommendations may warrant further comment and more specific detail may be needed for the work to be reproduced.

The methodology is sound and the work meets the expected standards for a Delphi consensus-based position paper.

Reviewer #2

(Remarks to the Author)

Thank you for the opportunity to review this paper. This document is highly needed and represents a comprehensive body of work. This document systematically summarizes important aspects of lipedema and gives insights in where the disagreements in the lipedema research fields are.

Introduction

Well-written introductions, which provides us the framework for understanding the rationale behind this consensus document in lipedema.

Aim

The aim of the document is clearly defined.

Methods

The methods section is well-written, and well-explained. A great and thorough job has been done.

Results

Statement 2: It is a bit unclear whether the statement means that pain is needed for lipedema diagnosis, or "just" discomfort. Does this mean that pain is now a recognized symptom and necessary for lipedema diagnosis? Or does this mean that discomfort is sufficient?

Statement 3: Could you please rephrase the argument in statement 3 a bit? It is somewhat unclear why the lack of definitions for anatomical regions is a problem. After references at page 7 lines 277.

Statement 5: It is somewhat unclear where the threshold for sufficient evidence lies. I understand that it is determined based on expert consensus, but as for the last sentences (lines 311 to 313), I see no reason to include them when the basis is so limited, and as you also stated as well, there is a lack of evidence. I understand that you included these sentences to explain why not everyone agrees, but I would reconsider whether they should be included.

Statement 7: Same here as the comment above. When it is a consensus document, arguments that are well-documented should be included. The last sentences do not have sufficient evidence, as you also point out. Please reconsider whether

they should be included.

Statement 8 and 10: Feelings of heaviness and easy bruising is also reported by Luta et al. 2025 (<https://pmc.ncbi.nlm.nih.gov/articles/PMC11925301/>)

Statement 15: I would be a bit more cautious about stating that the evidence 'collectively indicates,' given that there are studies showing no increase in ECF in lipedema (Stellmaker et al, 2024). Could it be rephrased a bit more cautiously? Perhaps remove 'collectively'?

Statement 20: I find the argumentation and discussion here somewhat unclear, and I do not fully understand where the disagreement lies. Could you please rephrase this paragraph?

Statement 21: I find the sentence about the adipose tissue between lipedema and obesity somewhat unclear (lines 562 to 564). Could you please rephrase it? For example, "the adipose tissue formed in concomitant lipedema and obesity..."

Statement 26: I find this comment largely characterized by speculation without sufficient evidence. Consider reducing it, or alternatively, mention that it is speculative. There is, moreover, a low degree of consensus, weak evidence, and agreement that more evidence is needed. However, still consider whether the sentence with "rusticanus" should be included.

Statement 29: In sentence 2, lines 702-705, you write that progression can be measured by pain and altered body image. I understand that body image can be affected by unsuccessful therapeutic interventions and ideas, but I suggest rephrasing this sentence. As it stands, I interpreted it as the progression of lipedema being measured by increased pain and altered body image. I would rather change it to an increase in leg volume, for example.

Statement 32: You may also consider including the study by Dinnendahl et al here as well, reference number 30.

Statement 33: I find it a bit difficult to understand where the disagreement lies. I see that the consensus is at 82%, but there is nothing in the comments that indicates anyone disagrees with this.

Statement 36: I find it a bit unclear what is meant in this sentence about the various factors contributing to differences in symptom progression. How would genetic predisposition affect differences in symptom progression?

Statement 37: Consider rephrasing limb volume excess to excess limb volume. Also, please rephrase the second sentence (lines 850-842). It seems a bit unclear where the disagreements are. Does this mean that some authors believe that lipedema can lead to obesity?

Statement 42: I find the meaning of the third sentence a bit unclear and suggest rephrasing it. What is meant by 'excessive strain,' and wouldn't walking be a weight-bearing activity that could lead to stress on muscles and joints? Could you also clarify muscle tone a bit? Additionally, I suggest choosing another reference that is available in English to make it easier for readers

Statement 45: Suggests adding this paper (<https://pubmed.ncbi.nlm.nih.gov/40125475/>) after effect of inflammation following ketogenic diets in lipedema. You can also consider adding "pain relief" in addition to symptom relief.

Statement 47: Please rephrase sentence number two (line 2035), as this sentence seem unclear. e.g. Obesity is usually classified using BMI (>30 kg/m²).

Statement 49: I suggest using an English-language reference instead of a German one, as it may make it easier for readers. I am also not sure that walking is associated with increased muscle strength, so I suggest rephrasing this sentence (line 1071-1073).

Statement 51: I find it a bit unclear how lipedema leads to the worsening of obesity symptoms, as referred to in your mention of 'vice versa.' What is classified as obesity symptoms, is it increased fat tissue or inflammation?

Discussion

I would like to emphasize the importance of detailing the distribution of patient representatives. Additionally, in lines 1287-1289 on page 29, you mention that the populations affected by lipedema remain underrepresented in this work. However, the table showing the distribution of patient representatives indicates a high percentage of the population. Could you clarify what is meant by this statement in the discussion? Are you referring to geographical distribution? Please also specify which populations, based on geography, are not included in the document.

Could you please clarify why there are so many more participants in round three compared to the earlier rounds?

Conclusion:

We understand that it is not possible to draw conclusions based on so many guidelines, but the conclusion feels more like a summary. If this is not determined by the journal, perhaps the heading could be changed to 'Summary'?

Acknowledgement

We see coauthors listed as "participated in the voting process, but not in the interpretation of the results". However, further down, it says that all coauthors were included in editing and commenting on the manuscript at all stages. We found it a bit unclear who participated in which parts. Could this be clarified?

Minor

Page 6, line 261: please remove the extra space between the references, should be phrased as (25-30)

Page 7, line 277: please remove the extra space between the references, should be phrased (31-37).

Page 7, line 296: typo, reports instead of report

Thanks for the opportunity to review this article.

Reviewer #3

(Remarks to the Author)

I have had the opportunity to review this Delphi study from a methodological point of view. The authors provide a well-reasoned rationale for the study design and have conducted the Delphi study with rigor and transparency. I offer the following suggestions to further improve the clarity of the methods.

Specific comments

1. The abstract uses the terms "90% to 100% consensus" and "80% to 90% consensus". I suggest replacing "consensus" with "agreement".
2. Line 176: "Round 2 included the aggregated results from Round 1". Please clarify what type of feedback was provided to participants (e.g. frequencies, means, summary of comments). Similarly, in line 193: "along with feedback from previous Delphi rounds" - it would be helpful to specify the type of this feedback.
3. Line 179f: "Five statements were removed from further evaluation due to an insufficient level of agreement." Please define what constitutes "insufficient level of agreement" and specify the threshold used.
4. You indicate that consensus was defined on the basis of criteria ii and iii. Please clarify the role of criteria i, iv and v. A brief explanation would enhance methodological transparency.

General comments

- I recommend using and referring to a Delphi reporting guideline, e.g. DELPHISTAR (via the EQUATOR network), to further enhance methodological transparency and comparability with other Delphi studies.
- Although my review focused on the methodology, the discussion could benefit from a clearer link to the existing literature to better contextualize the findings.

Reviewer #4

(Remarks to the Author)

Version 1:

Reviewer comments:

Reviewer #1

(Remarks to the Author)

The authors have incorporated most of the reviewers' feedback, and where they haven't, they provide clear reasons for disagreement. Certain statements have been reworded to allow for the reader's interpretation and highlight the varied opinions of the project team of experts.

Reviewer #2

(Remarks to the Author)

Reviewer #3

(Remarks to the Author)

I would like to thank the authors for their thoughtful and thorough revisions and detailed responses to the reviewers' comments. The manuscript is now clearer and more transparent. The authors have carefully addressed the comments. As all concerns have been addressed to my satisfaction, I support the publication of the manuscript in its current form.

Reviewer #4

(Remarks to the Author)

REVIEWER COMMENTS are noted in BLUE

AUTHORS' RESPONSES are noted in BLACK

REVIEWER COMMENTS

Reviewer #1 (Remarks to the Author):

The Delphi Consensus-Based Position Paper shows noteworthy consistent results and is a significant review of statements regarding the definition and management of lipoedema, building on the established literature. A comprehensive literature search justified the statements made, and the consensus highlights the need for further research in this condition. The process takes an inclusive approach and focuses on the diversity component. However, the data still lacks gaps to represent the spectrum of protected characteristics in the expert panel (such as gender and age).

Thank you for the supportive assessment of the manuscript's key elements. Instead of the participants' ages, the authors provided their years of experience in treating lipedema, as this is more relevant to the topic (Table 3). Gender information has been added in Section 3 (Results).

Line 224: *"In the final Delphi evaluation round 3, 71 out of 103 founding members of the LWA participated (34 female, 37 male), representing 19 countries across 5 continents (Table 2)."*

There were no obvious flaws in the data analysis, but some of the expert panel recommendations may warrant further comment and more specific detail may be needed for the work to be reproduced.

We thank the reviewer for the helpful input to improve the manuscript and refine specific wording. The expert panel comments were intentionally kept brief to support understanding, prevent misunderstandings, and provide context in cases of dissent. They are not intended as a comprehensive literature review, which would exceed the manuscript's scope. Nonetheless, we carefully reviewed all statements and revised the comments where appropriate. Please find additional information in responses to the comments from Reviewer 2.

The methodology is sound and the work meets the expected standards for a Delphi consensus-based position paper.

Once again, thank you for your positive evaluation of our methodology.

Reviewer #2 (Remarks to the Author):

Thank you for the opportunity to review this paper. This document is highly needed and represents a comprehensive body of work. This document systematically summarizes important aspects of lipedema and gives insights in where the disagreements in the lipedema research fields are.

Many thanks to Siren Nymo and Julianne Lundanes for recognizing the manuscript's importance and for their thorough, detailed analysis, which will significantly strengthen it.

Introduction

Well-written introductions, which provides us the framework for understanding the rationale behind this consensus document in lipedema.

Thank you.

Aim

The aim of the document is clearly defined.

Agreed.

Methods

The methods section is well-written, and well-explained. A great and thorough job has been done.

Thank you for your kind assessment.

Results

Statement 2: It is a bit unclear whether the statement means that pain is needed for lipedema diagnosis, or "just" discomfort. Does this mean that pain is now a recognized symptom and necessary for lipedema diagnosis? Or does this mean that discomfort is sufficient?

The formulation of this statement required extensive deliberation by the authors and expert panel, due to the wide international variation in how this symptom is described. The second part of the expert panel comments serves to clarify the selected phrasing, noting: *„The presence of pain varies widely among individuals, ranging from mild discomfort to severe, chronic pain.“* Statement 6 provides additional detail on pain in lipedema and emphasizes the pressing need for a standardized definition.

Statement 3: Could you please rephrase the argument in statement 3 a bit? It is somewhat unclear why the lack of definitions for anatomical regions is a problem. After references at page 7 lines 277.

The distinction between torso and extremities shapes the debate on which regions are affected by lipedema, given that the torso is generally seen as unaffected. It also may assist in treatment planning of affected regions to best understand their anatomical extent. For example, the pelvic girdle, though positioned in the lower torso, is commonly classified as part of the extremities based on embryological, functional and evolutionary considerations. We agree that the challenge stems from inconsistent terminology rather than a lack of clear definitions. The section has been adjusted to reflect this.

Line 289: *“However, inconsistent terminology for anatomical regions in the literature necessitates clarification in order to more clearly discuss the regions affected by lipedema.”*

Statement 5: It is somewhat unclear where the threshold for sufficient evidence lies. I understand that it is determined based on expert consensus, but as for the last sentences (lines 311 to 313), I see no reason to include them when the basis is so limited, and as you also stated as well, there is a lack of evidence. I understand that you included these sentences to explain why not everyone agrees, but I would reconsider whether they should be included.

We appreciate the ability to recognize and address internal discussions within the expert panel and author team during the review process. Ultimately, the authors agreed that the expert panel comment should transparently address the dissent concerning the exclusion of hands and feet in lipedema SAT accumulation. The statement about absence of definitive evidence potentially serves as a call for research about this aspect of lipedema, and case examples are important to note as

initial evidence. The etiology is left open, linking it to disease progression or in relation to possible comorbidities. Overall, this clarification is considered important to explain why near-unanimous agreement on such a clear statement was not achieved. Therefore, we respectfully wish to keep the present discussion rephrasing to:

Line 324: *"However, although not constituting definitive evidence, patient reports suggest [...]"*

Statement 7: Same here as the comment above. When it is a consensus document, arguments that are well-documented should be included. The last sentences do not have sufficient evidence, as you also point out. Please reconsider whether they should be included.

The authors aim to clarify that symptoms outside the extremities, excluding hands and feet, are likely related to comorbidities. Without this specification, the approximately 85% agreement rate might be difficult to interpret. We have rephrased the statement about evidence to:

Line 358: *"However, although not constituting definitive evidence, patient reports suggest [...]"*

Statement 8 and 10: Feelings of heaviness and easy bruising is also reported by Luta et al. 2025 (<https://pmc.ncbi.nlm.nih.gov/articles/PMC11925301/>)

Thank you for recommending this reference. It has now been incorporated, as it was not accessible when the manuscript was first submitted.

Statement 15: I would be a bit more cautious about stating that the evidence 'collectively indicates,' given that there are studies showing no increase in ECF in lipedema (Stellmaker et al, 2024). Could it be rephrased a bit more cautiously? Perhaps remove 'collectively'?

We acknowledge that the original phrasing may have been overly definitive and have removed "collectively" as recommended. Additionally, the report by Stellmaker et al was not available during the Delphi process, and we will therefore not include it, given its potential to impact the statement. Notably, the study excluded participants with lymphatic insufficiency on ICG-lymphography, which may explain the absence of ECF in their cohort.

Statement 20: I find the argumentation and discussion here somewhat unclear, and I do not fully understand where the disagreement lies. Could you please rephrase this paragraph?

The differing assessments were made to highlight that BMI is an inadequate measure for differentiating lipedema from obesity. This aspect was further emphasized in the panel comment:

Line 555: *"[...] However, BMI is not an ideal tool to assess obesity in patients with lipedema due to the disproportionate body habitus inherent in lipedema [99], potentially contributing to divergent clinical evaluations and interpretations of existing literature. Nevertheless, when both conditions coexist, appropriate treatment must be provided for each as a separate disease. [...]"*

Statement 21: I find the sentence about the adipose tissue between lipedema and obesity somewhat unclear (lines 562 to 564). Could you please rephrase it? For example, "the adipose tissue formed in concomitant lipedema and obesity..."

The section was rephrased to improve clarity.

Line 577: *"Lipedema adipose tissue exhibits distinct morphological, molecular and metabolic characteristics compared to obesity-type adipose tissue [38, 104]."*

Statement 26: I find this comment largely characterized by speculation without sufficient evidence. Consider reducing it, or alternatively, mention that it is speculative. There is, moreover, a low degree of consensus, weak evidence, and agreement that more evidence is needed. However, still consider whether the sentence with "rusticanus" should be included.

The Reviewer's assessment of this statement as speculative is reflected in the lower agreement among the panel for this statement. The current supporting evidence and clinical observations are important to keep because this represents that basis of the statement. Recognizing the speculative nature of the connection between lipedema and HSD, the authors have revised the expert panel comments for clarity. The mention of the sub-type "rusticanus Moncorps type" remains, as it presents a potentially testable hypothesis.

Line 661: *"Evidence suggests that women with lipedema exhibit decreased elasticity in both skin [119] and aorta [120], along with joint hypermobility [22, 121, 122] and muscle weakness [123]. There is speculation about a link between lipedema and connective tissue disorders, specifically hypermobility spectrum disorders (HSD), although no conclusive underlying mechanism has been identified [124]. Considering the inconsistency of HSD in lipedema, a hypothesis has proposed categorizing it as a subtype ("rusticanus Moncorps type") of lipedema [119]. Further investigation is needed to confirm a definitive association between lipedema and HSD."*

Statement 29: In sentence 2, lines 702-705, you write that progression can be measured by pain and altered body image. I understand that body image can be affected by unsuccessful therapeutic interventions and ideas, but I suggest rephrasing this sentence. As it stands, I interpreted it as the progression of lipedema being measured by increased pain and altered body image. I would rather change it to an increase in leg volume, for example.

Thank you for highlighting the potential for misunderstanding. The phrase "altered body image" has been revised to "disproportionate adipose tissue accumulation" to more precisely describe the clinical progression.

Statement 32: You may also consider including the study by Dinnendahl et al here as well, reference number 30.

This reference should clearly be included here, as the method holds great potential to address the earlier-discussed ambiguity in categorizing the symptom "pain." The source has been added. Thank you for this suggestion.

Statement 33: I find it a bit difficult to understand where the disagreement lies. I see that the consensus is at 82%, but there is nothing in the comments that indicates anyone disagrees with this.

The main point of disagreement was that some experts wished to eliminate BMI as an anthropometric measure for patients with lipedema. A statement was included to reflect this divergence and to emphasize the authors' view that BMI remains an important standard measure.

Line 791: *"Despite its limited significance in lipedema, BMI can support disease monitoring due to its simplicity in measurement and interpretation."*

Statement 36: I find it a bit unclear what is meant in this sentence about the various factors contributing to differences in symptom progression. How would genetic predisposition affect differences in symptom progression?

The central point of this statement is that disease progression (e.g., by clinical stages) does not inevitably occur, unlike in lymphedema. Although progression is frequently seen, its pattern is quite heterogeneous. This is adequately clarified in the first section of the expert panel commentary. The second part refers to factors potentially involved, including hormonal changes, weight gain, and lifestyle influences, which are undisputed contributors. We agree that "genetic predisposition" may not have been the most precise term and have revised it to "individual predisposition." We also agree that robust evidence exists mainly for weight gain, as noted in the concluding sentence.

Statement 37: Consider rephrasing limb volume excess to excess limb volume. Also, please rephrase the second sentence (lines 850-842). It seems a bit unclear where the disagreements are. Does this mean that some authors believe that lipedema can lead to obesity?

The divergence of opinion relates to patients affected by both lipedema and obesity. It was clarified that, in such cases, a clear distinction cannot yet be made, and both conditions may contribute to excess limb volume.

Line 862: *"Given the high number of patients with lipedema who also suffer from concomitant obesity, excess limb volume can be attributed to both underlying conditions."*

Statement 42: I find the meaning of the third sentence a bit unclear and suggest rephrasing it. What is meant by 'excessive strain,' and wouldn't walking be a weight-bearing activity that could lead to stress on muscles and joints? Could you also clarify muscle tone a bit? Additionally, I suggest choosing another reference that is available in English to make it easier for readers

We recognize that the original phrasing did not adequately reflect the evidence level and have adjusted the section. "Aquatic exercise" now replaces "swimming or walking," and "muscle strength" replaces "muscle tone." Relevant English-language references have been added.

Line 958: *"Tailored exercise regimes, such as aquatic exercise, are often recommended to promote lymphatic flow [150] and muscle strength without placing excessive strain on affected limbs [149, 151]."*

Statement 45: Suggests adding this paper (<https://pubmed.ncbi.nlm.nih.gov/40125475/>) after effect of inflammation following ketogenic diets in lipedema. You can also consider adding "pain relief" in addition to symptom relief.

The reference has been included. In light of the unclear definition of lipedema pain and the variability in symptom presentation, the authors prefer to refer generally to 'symptom relief' from ketogenic diets rather than specifically mentioning 'pain'.

Statement 47: Please rephrase sentence number two (line 2035), as this sentence seems unclear. e.g. Obesity is usually classified using BMI (>30 kg/m²).

The sentence was rephrased to improve clarity.

Line 1055: *"Obesity classification typically relies on BMI. However, in the context of lipedema and in line with statement 33, utilizing alternative anthropometric measures—such as WHtR—may be more appropriate for assessing or excluding*

overweight and obesity, particularly in the context of metabolic health categorization [100]."

Statement 49: I suggest using an English-language reference instead of a German one, as it may make it easier for readers. I am also not sure that walking is associated with increased muscle strength, so I suggest rephrasing this sentence (line 1071-1073).

English-language references were added where necessary and appropriate. The sentence, which links yoga and walking to increased muscle strength, was revised to present them as examples of "structured exercise." Additionally, a reference was included that the authors believe sufficiently supports the association between long-term walking exercise and increased muscle strength.

Line 1092: *"Moreover, structured exercise—such as walking and yoga—have been associated with improved muscular strength, and psychological well-being, factors that are particularly relevant for individuals with lipedema [123, 178, 179]."*

Statement 51: I find it a bit unclear how lipedema leads to the worsening of obesity symptoms, as referred to in your mention of 'vice versa.' What is classified as obesity symptoms, is it increased fat tissue or inflammation?

The sentence was revised to clarify causality. Lipedema-related symptoms can significantly impair general mobility—e.g., due to excess tissue causing mechanical irritation and movement-related pain—suggesting that reduced physical activity may contribute to the progression of coexisting obesity. Furthermore, patients with both lipedema and obesity tend to experience significantly diminished quality of life, which may reduce motivation for physical activity and the adoption of a healthy lifestyle. Additional references have been added.

It should also be noted that the authors refer to a worsening of "obesity" in general, not specifically to "obesity symptoms," thereby including comorbidities such as type 2 diabetes, hypertension, dyslipidemia, nonalcoholic fatty liver disease, and polycystic ovary syndrome.

Line 1120: *"Existing literature highlights the bidirectional interplay between obesity and lipedema, with obesity potentially exacerbating lipedema symptoms [183] and vice versa due to impaired mobility, pain or psychological burden linked to lipedema [19, 46, 175, 177, 184]."*

Discussion

I would like to emphasize the importance of detailing the distribution of patient representatives.

A section was added at the beginning of the results to address the geographic distribution of patient advocates in the third Delphi round.

Line 230: *"Seven countries in total were represented by the patient advocates, with ten of the twelve originating from Europe."*

Additionally, in lines 1287-1289 on page 29, you mention that the populations affected by lipedema remain underrepresented in this work. However, the table showing the distribution of patient representatives indicates a high percentage of the population. Could you clarify what is meant by this statement in the discussion? Are you referring to geographical distribution? Please also specify which populations, based on geography, are not included in the document.

As correctly assumed, this clause refers to the geographic distribution of participants in the Delphi process. There are no participants from Africa, only one

from India in Asia, two from Brazil in South America, and two from Australia. Given the predominance of participants from Europe and the United States, several significant populations are notably underrepresented.

Line 1309: *"Additionally, although experts from 19 countries participated, significant populations affected by lipedema—such as those in Asia and Africa—remain underrepresented in this work."*

Could you please clarify why there are so many more participants in round three compared to the earlier rounds?

This section was included following the relevant remark in the study limitations to provide clarification:

Line 1373: *"The in-person meeting after Round 2 during the dedicated session at the Lipedema World Congress in Potsdam on October 7, 2023, significantly increased the project's visibility and attention. This led to a higher number of participants in the third Delphi round, ensuring the final results are well-founded and reliable."*

Conclusion:

We understand that it is not possible to draw conclusions based on so many guidelines, but the conclusion feels more like a summary. If this is not determined by the journal, perhaps the heading could be changed to 'Summary'?

The authors have no preference and respectfully defer to the Editorial Board to decide whether "Summary" would be more appropriate.

Acknowledgement

We see coauthors listed as "participated in the voting process, but not in the interpretation of the results". However, further down, it says that all coauthors were included in editing and commenting on the manuscript at all stages. We found it a bit unclear who participated in which parts. Could this be clarified?

It was clarified that all listed authors also participated in the Delphi process, whereas those mentioned in the acknowledgments voted but did not participate in interpreting the findings. Additionally, the authors' names have been removed from the acknowledgments in the Delphi voting process.

Line 2003: *"In addition to the author team, the below specialists participated in the voting process of the Delphi methodology, but not in interpretation of the results, and agreed to be mentioned here."*

Minor

Page 6, line 261: please remove the extra space between the references, should be phrased as (25-30)

The space has been removed.

Page 7, line 277: please remove the extra space between the references, should be phrased (31-37).

The space has been removed.

Page 7, line 296: typo, reports instead of report

The typo has been corrected.

Thanks for the opportunity to review this article.

Reviewer #3 (Remarks to the Author):

I have had the opportunity to review this Delphi study from a methodological point of view. The authors provide a well-reasoned rationale for the study design and have conducted the Delphi study with rigor and transparency. I offer the following suggestions to further improve the clarity of the methods.

Thank you for recognizing our commitment to working accurately and transparently.

Specific comments

1. The abstract uses the terms "90% to 100% consensus" and "80% to 90% consensus". I suggest replacing "consensus" with "agreement".

The comment was well received, and the wording was adjusted accordingly. The abstract now states:

Line 68: *"Thirty-six statements reached 90% to 100% agreement, 17 statements reached 80% to 90% agreement, and 6 statements reached 70% agreement."*

2. Line 176: "Round 2 included the aggregated results from Round 1". Please clarify what type of feedback was provided to participants (e.g. frequencies, means, summary of comments).

During this phase of the Delphi voting, participants were provided with a summary of comments when necessary to understand changes from the previous version. Approval rates were not provided.

Line 190: *"Participants were provided with a summary of comments when necessary to understand changes from the previous version."*

After the second survey round, relevant comments were discussed at the on-site meeting, and the percentage of votes scoring ≥ 4 on the five-point Likert scale or the percentage of yes responses for each statement was disclosed. Additionally, in the third online Delphi round, participants were shown the expert panel comment formulated based on the feedback.

Line 195: *"During the on-site meeting, results showing the percentage of votes scoring ≥ 4 on the five-point Likert scale or the percentage of yes responses per statement were shared and related comments were reviewed."*

Similarly, in line 193: "along with feedback from previous Delphi rounds" - it would be helpful to specify the type of this feedback.

This refers to related open comments on statements with limited agreement rates that required clarification.

Line 203: *"[...] along with open-ended feedback from previous Delphi rounds [...]"*

3. Line 179f: "Five statements were removed from further evaluation due to an insufficient level of agreement." Please define what constitutes "insufficient level of agreement" and specify the threshold used.

Consensus was previously defined as greater than 70% expert agreement on a criterion, accompanied by more than 70% of participants recommending its

inclusion in the document. In this context, this is equivalent to "sufficient agreement." This clarification has been added.

Line 161: "*Consensus, or sufficient agreement, for a given statement was defined a priori as greater than 70% of expert agreement on criterion (ii), along with more than 70% of participants recommending inclusion in the document on criterion (iii) [12-15].*"

4. You indicate that consensus was defined on the basis of criteria ii and iii. Please clarify the role of criteria i, iv and v. A brief explanation would enhance methodological transparency.

A section addressing the relevance of Criteria (i), (iv), and (v) has been added.

Line 164: "*Criteria (i), (iv), and (v) are not relevant to the consensus threshold but serve to contextualize each statement.*"

General comments

- I recommend using and referring to a Delphi reporting guideline, e.g. DELPHISTAR (via the EQUATOR network), to further enhance methodological transparency and comparability with other Delphi studies.

The DELPHISTAR reporting guidelines were applied, with the filled checklist provided in the supplementary files.

Line 128: "*The study was conducted and reported following the DELPHISTAR reporting guidelines for Delphi studies in health research [12]. A completed DELPHISTAR checklist is provided in supplemental file xxx.*"

- Although my review focused on the methodology, the discussion could benefit from a clearer link to the existing literature to better contextualize the findings.

The authors respectfully disagree with the reviewer on this point. The manuscript is intentionally structured so that each statement is accompanied not only by an objective evaluation but also by an "expert panel comment" placing it in the context of existing literature. Providing a full contextualization of all findings would exceed the scope of the discussion, which is why a different structure was chosen from the outset. With due respect to the reviewer's comment, the authors prefer to maintain the current manuscript format.

Reviewer #4 (Remarks to the Author):

We sincerely appreciate your involvement as an Early Career Researcher, and the thorough review you provided of our manuscript. The valuable feedback has clearly enhanced its quality.